# Unified Gradient-Based Machine Unlearning with Remain Geometry Enhancement

**Zhehao Huang, Xinwen Cheng, JingHao Zheng, Haoran Wang, Zhengbao He, Tao Li, Xiaolin Huang**
Shanghai Jiao Tong University
[kinght_H, xinwencheng, zjh20030406, haoran_whynot, lstefanie, li.tao, xiaolinhuang]@sjtu.edu.cn

## Abstract

Machine unlearning (MU) has emerged to enhance the privacy and trustworthiness of deep neural networks. Approximate MU is a practical method for large-scale models. Our investigation into approximate MU starts with identifying the steepest descent direction, minimizing the output Kullback-Leibler divergence to exact MU inside a parameters' neighborhood. This probed direction decomposes into three components: weighted forgetting gradient ascent, fine-tuning retaining gradient descent, and a weight saliency matrix. Such decomposition derived from Euclidean metric encompasses most existing gradient-based MU methods. Nevertheless, adhering to Euclidean space may result in sub-optimal iterative trajectories due to the overlooked geometric structure of the output probability space. We suggest embedding the unlearning update into a manifold rendered by the remaining geometry, incorporating second-order Hessian from the remaining data. It helps prevent effective unlearning from interfering with the retained performance. However, computing the second-order Hessian for large-scale models is intractable. To efficiently leverage the benefits of Hessian modulation, we propose a fast-slow parameter update strategy to implicitly approximate the up-to-date salient unlearning direction. Free from specific modal constraints, our approach is adaptable across computer vision unlearning tasks, including classification and generation. Extensive experiments validate our efficacy and efficiency. Notably, our method successfully performs class-forgetting on ImageNet using DiT and forgets a class on CIFAR-10 using DDPM in just 50 steps, compared to thousands of steps required by previous methods. Code is available at Unified-Unlearning-w-Remain-Geometry.

## 1 Introduction

Machine Unlearning (MU) [1–3] aims to remove the influence of samples from a pre-trained model, ensuring the model behaves as if it has never encountered those samples. The significance of MU research has grown following data protection regulations [4, 5]. It has rapidly developed in recent years, becoming an important means to help pre-trained large-scale models adapt to various trustworthy challenges [6] in computer vision (CV). In general, MU aids in purging outdated knowledge [7, 8], mitigating biases [9, 10], and preventing large text-to-image models from generating not-safe-for-work (NSFW) images [11–14].

Existing MU methods are mainly divided into two categories: *exact (or certified) MU* [15] and *approximate MU* [16]. For deep neural networks, achieving exact MU necessitates retraining on a dataset excluding forgetting samples. However, retraining is computationally prohibitive for recent large-scale deep networks. Thus, in MU for deep networks, this retrained model only serves as an aspirational standard to be approached [17]. We primarily focus on more efficient approximate MU.

Approximate MU strives to align the output distribution of unlearned models with that of retrained models. We initially explore the vanilla gradient descent of minimizing the output Kullback-Leibler

38th Conference on Neural Information Processing Systems (NeurIPS 2024).

(KL) divergence between the current iteration and the retrained model. Our deduction reveals that this direction consists of three parts: a weighted gradient ascent to eliminate the influence of forgetting samples, a descent gradient for fine-tuning the remaining set, and a weight saliency matrix modulating the unlearning direction. Such decomposition provides a novel perspective that unifies previous MU approaches proposed in recent years [18–25], most of which only focus on one or two of these components. For example, SalUn [26] insightfully proposes saliency-based unlearning, which only optimizes the parameters important to forget, but lacks theoretical support. Our analysis fills this gap and provides new directions for further improvement.

In fact, the vanilla gradient descent for approximate MU actually pursues the steepest descent under *Euclidean distance* [27–30]. However, constraining parameter updates within an Euclidean region is arbitrary, as it treats the importance of all parameters equally. Recent research indicates deep models' parameters and training processes are embedded in a low-dimensional manifold [31, 32]. Thus, there exist manifolds where changes in Euclidean between parameters are drastic, yet the output space remains unchanged. In MU, it is evident that the importance of model parameters varies for forgetting and remaining [33, 26], prompting the following question. *Can model parameters be embedded in a manifold that allows effective forgetting while efficiently maintaining remaining performance?*

To achieve this goal, we propose to discover the descent direction under the KL divergence on the remaining output distribution. Using such a manifold metric, the forgetting direction can be amended by a second-order Hessian on the remaining set to prevent forgetting loss from harming retained performance. Such iterative direction is dominated by the unlearning function, allowing the optimization process to focus on efficient forgetting. However, computing the second-order Hessian for large-scale models is computationally intensive, contradicting the need for efficiency in unlearning. Existing methods for estimating the second-order Hessian rely on initial parameters and keep them fixed thereafter [34, 35]. Therefore, we propose a fast-slow weight [36, 37] method (**Fig. 1**) to implicitly and dynamically approximate the salient forgetting direction with Hessian modulation, forming a unified MU approach for CV unlearning tasks including image classification and image generation. Key **contributions** of this paper include:

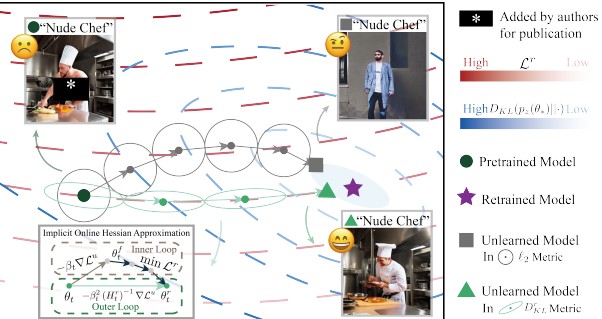

Figure 1: Overview of our proposal vs. previous unlearning methods on erasing concept 'nudity' in diffusion models [11, 12]. Conventional methods seek the steepest descent within an Euclidean ball, often compromising general capabilities. In contrast, we reach the region around retraining along a remain-preserving manifold. To address the large cost of Hessian, we implicitly approximate the up-to-date salient unlearning direction.

• We provide a novel perspective to unify previous approaches by decomposing the vanilla gradient descent direction of approximate MU into three components: weighted forgetting gradient ascent, remaining gradient descent, and a weight saliency matrix.

• We derive the steepest descent direction for approximate MU on the remain-preserved manifold.

• We propose a fast-slow weight method to implicitly approximate online Hessian-modulated salient forgetting updates.

• We conduct experiments on a wide range of CV unlearning tasks across multiple datasets and models of different architectures, verifying the effectiveness and efficiency of our method.

## 2 Preliminary

**Problem Setup.** MU aims to help a well-trained model eliminate the influence of specific data points, categories, or high-level concepts and patterns [1, 38]. Let $\mathcal{D} = \{z_i\}_{i=1}^{N}$ represent a pretraining dataset of $N$ data points, including $z_i = (x_i, y_i)$ features and labels in supervised learning. The *forgetting dataset*, $\mathcal{D}^f = \{z_i^f\}_{i=1}^{N^f} \subset \mathcal{D}$, is a subset of the pretrained dataset. Its complement, $\mathcal{D}^r = \{z_i^r\}_{i=1}^{N^r} = \mathcal{D}\backslash\mathcal{D}^f$, is the *remaining dataset* that we wish to retain. The learner is a model parameterized by $\theta$. $p_z(\theta) = p(z; \theta)$ represents the model output probability. The pre-trained model

is obtained by empirical risk minimization, i.e., $\theta_0 = \arg\min_\theta \mathcal{L}(\theta; \mathcal{D}) = \arg\min_\theta \sum_{i \in \mathcal{D}} \ell(\theta; z_i)$, as **Pretrain**. This empirical risk can be divided into two parts based on the forgetting and retaining datasets. $\mathcal{L}(\theta; \mathcal{D}) = \sum_{i \in \mathcal{D}^f} \varepsilon_i \ell(\theta; z_i^f) + \sum_{j \in \mathcal{D}^r} \ell(\theta; z_j^r) = \mathcal{L}^f(\theta; \varepsilon) + \mathcal{L}^r(\theta)$, where $\varepsilon = \{\varepsilon_i\}_{i=1}^{N_f}$ weight the former part [39, 38]. For the pre-trained model, the coefficients $\varepsilon_0 = 1$ are all ones. The following are the instantiations in CV unlearning tasks. In classification (Cls) problems, models output the class posterior probability $p(z_{\text{Cls}}; \theta) = p(y|x; \theta)$ and the empirical risk for each sample is the cross-entropy (CE) loss $\ell_{\text{Cls}}(\theta; z) = \ell_{\text{CE}}(\theta; x, y)$. In the image generation (Gen) task of conditional diffusion models [40–42], the output is the conditional sampling probability $p(z_{\text{Gen}}; \theta) = p(x|y; \theta)$, and the average loss function for each sample is the mean squared error (MSE) loss: $\ell_{\text{Gen}}(\theta; z) = \ell_{\text{MSE}}(\theta; x, y) = \mathbb{E}_{t, \epsilon \sim \mathcal{N}(0,1)}[\|\epsilon - \epsilon_\theta(x_t|y)\|_2^2]$, where $t$ represents the diffusion step, $\epsilon$ is the random noise sampled from a Gaussian distribution $\mathcal{N}(0, 1)$, $\epsilon_\theta$ is the conditional denoising model, and $x_t$ is the noisy version of the input image. For a more detailed introduction to image generation using conditional diffusion models and latent diffusion models, please refer to Appendix B.3.

**Exact and Approximate Machine Unlearning.** *Exact MU* [15, 43, 1] ensures that the parameter distribution of the unlearned model is identical to that of a model trained from scratch without seeing the forgetting samples. For large-scale models, exact unlearning can only be achieved by retraining (**RT**) on the remaining dataset $\theta_* = \arg\min_\theta \mathcal{L}^r(\theta)$. However, the computational cost of retraining in response to every forgetting request is prohibitive. Therefore, we regard RT as *a gold standard* to approximate rather than a competitor. A more practical approach is to guide the unlearned model output distribution to approximate the output distribution of RT, known as *approximate MU* [16]. If we use KL divergence to measure the difference in output distributions, the objective of approximate unlearning can be expressed as: $\min_\theta D_{\text{KL}}(p_z(\theta_*)||p_z(\theta)) = \min_\theta \int p_z(\theta_*) \log[p_z(\theta_*)/p_z(\theta)]\mathrm{d}\mathcal{D}$, starting from $\theta_0$. Therefore, a straight metric to approximate unlearning is the KL divergence from the retrained model's output distribution. In addition, we also investigate metrics related to forgetting efficacy, retained performance, and privacy protection for evaluation in image classification and generation tasks. For details of evaluation, please refer to **Sec. 5** and Appendix D.

**Steepest Descent.** Approximate MU methods are usually based on gradient updates to obtain the unlearned model [18, 21]. Let's first reinterpret the gradient with *steepest descent* [29, 27, 30]. The goal of steepest descent is to find the direction $\delta\theta = \theta_{t+1} - \theta_t$ that drives the objective function $F(\theta)$ descent fastest within a $\xi$-neighborhood of the current parameters $\theta_t$. This can be formulated as the following optimization problem. (See Appendix A.1 for proof.)

$$\delta\theta := \underset{\rho(\theta_t, \theta_t + \delta\theta) \leq \xi}{\arg\min} F(\theta_t + \delta\theta) \Rightarrow \theta_{t+1} := \underset{\theta_{t+1}}{\arg\min} F(\theta_{t+1}) + \frac{1}{\alpha_t(\xi, \theta_t)}\rho(\theta_t, \theta_{t+1}), \quad (1)$$

where $\rho(\cdot, \cdot)$ represents the manifold metric that renders the geometry of the parameter's neighborhood. To simplify the derivation, we rewrite it to the form on the right, where $\alpha_t(\xi, \theta_t)$ represents the learning rate required to move the distance $\xi$. In the following, we fix a small learning rate $\alpha_t$ to approximate a search within a local neighborhood. The characterization $\rho$ of the underlying coordinate space of the neighborhood will determine update directions, optimization paths, and the flatness of the minimum. Vanilla gradient descent is obtained using the Euclidean metric, Newton's direction is measured by the second-order expansion of the objective function [44], and the KL divergence in the output space induces a natural gradient [30, 45]. Next, we will probe approximate MU through vanilla gradient descent and attempt to benefit from improved manifold metrics.

## 3 Approximate MU from Perspective of Steepest Descent

**Revisit Approximate MU Methods via Vanilla Gradient Descent.** We begin with the vanilla gradient descent direction to address approximate MU. This involves finding the steepest descent direction that minimizes the KL divergence with the retrained output within the vicinity of the current model $\theta_t$. The optimization problem can be formalized as follows:

$$\theta_{t+1} = \underset{\theta_{t+1}}{\arg\min} D_{\text{KL}}(p_z(\theta_*)||p_z(\theta_{t+1})) + \frac{1}{\alpha_t}\rho(\theta_t, \theta_{t+1}) \tag{2}$$

$$= \underset{\theta_{t+1}}{\arg\min} \underbrace{D_{\text{KL}}(p_{z^f}(\theta_*)||p_{z^f}(\theta_{t+1}))}_{(a)} p^f + \underbrace{D_{\text{KL}}(p_{z^r}(\theta_*)||p_{z^r}(\theta_{t+1}))}_{(b)} p^r + \frac{1}{\alpha_t}\underbrace{\rho(\theta_t, \theta_{t+1})}_{(c)},$$

where $p^f = p(\mathcal{D}^f|\mathcal{D})$ and $p^r = p(\mathcal{D}^r|\mathcal{D}) = 1 - p_f$ denote the partition of forgetting and remaining dataset, respectively. Analyzing (2), $(a)$ seeks to eliminate the influence of the target forgetting

samples, (*b*) aims to maintain the performance on the remaining samples, and (*c*) employs the metric $\rho$ to constrain the magnitude of each update, thereby identifying the direction of steepest descent on the manifold. To solve the optimization challenge outlined in (2), we posit that for the current model $\theta_t$, there exists a set of coefficients $\varepsilon_t = \{\varepsilon_{t,i}\}_{i=1}^{N_f}$ that weights the forgetting loss, positioning $\theta_t$ as the minimizer of the weighted loss for the original training set. The unlearning process necessitates adaptations in coefficients of forgetting loss. Then, we can determine the vanilla gradient descent for approximate MU by using *Euclidean distance $\ell_2$* as the manifold metric, as stated in **Prop. 1**.

**Proposition 1.** *Under the Euclidean manifold metric, $\rho(\theta_t, \theta_{t+1}) = \frac{1}{2}\|\theta_t - \theta_{t+1}\|^2$. Assuming that the current model $\theta_t = \arg\min_\theta \mathcal{L}^f(\theta; \varepsilon_t) + \mathcal{L}^r(\theta)$. Let $H_*^f = \nabla^2 \mathcal{L}^f(\theta_*; \mathbf{1})$ and $H_*^r = \nabla^2 \mathcal{L}^r(\theta_*)$ denote the Hessian of the retrained model on the forgetting set and the remaining set, respectively. Then, the steepest descent direction that minimizes (2) is approximately:*

$$\theta_{t+1} - \theta_t :\approx -\alpha_t [\underbrace{H_*^f (H_*^r)^{-1}}_{(S)} \underbrace{[-\nabla \mathcal{L}^f(\theta_t; \varepsilon_t)]}_{(F)} p^f + \underbrace{\nabla \mathcal{L}^r(\theta_t)}_{(R)} p^r]. \tag{3}$$

The proof can be found in Appendix A.2. To elucidate the effectiveness of gradient-based unlearning methods, we decompose the vanilla gradient descent direction in (3) into three components: (F), (R), and (S). (F) represents the gradient ascent direction of the weighted forgetting loss, which directs the model to discard the information of the forgetting samples. Fine-tuning (**FT**) [22, 19, 38] fails to guarantee MU due to the absence of (F). Current approximate MU methods such as Random Labeling (**RL**) [20] and BadTeacher knowledge distillation (**BT**) [23] are akin to weighted forgetting

Table 1: Comparison of approximate MU methods. We decompose the steepest descent direction into three parts: the weight saliency matrix (S), the forgetting part (F), and the remaining part (R) as in (3) and (4). Only SA and our method consider the remain-preserving manifold, and we further approximate up-to-date Hessian.

| Approximate MU Methods | Task | | MU components | | | Manifold Metric | Online Hessian |
|---|---|---|---|---|---|---|---|
| | Cls | Gen | (S) | (F) | (R) | | |
| FT [22, 19, 38] | ✓ | ✓ | | | ✓ | $\ell_2$ | |
| GA [20, 21] | ✓ | ✓ | | ✓ | | $\ell_2$ | |
| BT [23] | ✓ | | | ✓ | ✓ | $\ell_2$ | |
| SalUn [26] | ✓ | ✓ | ✓ | ✓ | ✓ | $\ell_2$ | |
| SA [35] | | ✓ | | ✓ | ✓ | $D_{\mathrm{KL}}^r$ | |
| SFR-on | ✓ | ✓ | ✓ | ✓ | ✓ | $D_{\mathrm{KL}}^r$ | ✓ |

loss gradient ascent, uniformly leading to an increase in the loss on forgetting samples. The unlearning process often causes catastrophic forgetting of the retained knowledge. Thus, it is common to integrate (R) fine-tuning on the remaining set to sustain the model's general capabilities. Unlearned models via Gradient Ascent (**GA**) [20, 21] usually lose usability without (R). Furthermore, (S), ignored in most of the previous literature, involves two Hessian modulation parts. $H_*^f$ amplifies the parameter updates crucial for forgetting, while $(H_*^r)^{-1}$ dampens those important for maintaining. The notion of (S) closely mirrors the *Weight Saliency* introduced in **SalUn** [26]. We provide theoretical support for this notion. Importantly, our framework makes no assumption regarding input modalities, allowing its flexible application across various CV unlearning tasks.

**Approximate MU in Remain-preserving Manifold.** In fact, employing Euclidean distance as the manifold metric for parameter updates is arbitrary. It treats all coordinates as equally important because the local second-order expansion is identical $\nabla_{\theta_t}^2 (\frac{1}{2}\|\theta_t - \theta_{t+1}\|^2) = I$. This uniform treatment overlooks the varying parameter significance for forgetting and remaining. Moreover, certain manifolds of parameter space can exhibit substantial variations in Euclidean metric, yet the induced model output remains almost unchanged [46]. Since the retrained model performance on forgetting is unpredictable, it is pragmatic to introduce manifolds related to the remaining. Therefore, a practical objective is to constrain parameter updates during unlearning within a manifold that minimally impacts the retained performance. An empirical characterization of such a manifold could be *the KL divergence on the output distribution of the remaining set*, $D_{\mathrm{KL}}^r$. Given that the original well-trained and the retrained model output closely match the ground-truth remaining distribution, $\nabla \mathcal{L}^r(\theta_0) \approx \nabla \mathcal{L}^r(\theta_*) \approx 0$. By starting with $\theta_0$ and limiting updates to this manifold, the maintained output distribution remains almost consistent throughout the unlearning iterations, $\nabla \mathcal{L}^r(\theta_{t+1}) \approx \nabla \mathcal{L}^r(\theta_t) \approx \nabla \mathcal{L}^r(\theta_0) \approx 0$. This consistency permits a second-order Taylor expansion at $\theta_t$ to terms (b) and (c) in (2), providing crucial curvature information for unlearning to prevent deviations in the model output on the remaining set, leading to **Prop. 2**.

**Proposition 2.** *Using the model output KL divergence on the remaining set as the manifold metric, $\rho(\theta_t, \theta_{t+1}) = D_{KL}(p_{z^r}(\theta_t)\|p_{z^r}(\theta_{t+1}))$. Assuming that the current model $\theta_t = \arg\min_\theta \mathcal{L}^r(\theta) + \mathcal{L}^f(\theta; \varepsilon_t)$. Let $\tilde{\alpha}_t = \alpha_t p^f / (\alpha_t p^r + 1)$, and $H_t^r = \nabla^2 \mathcal{L}^r(\theta_t)$ represent the Hessian w.r.t. $\theta_t$ on the remaining set, then the steepest descent direction that minimizes (2) is approximately:*

$$\theta_{t+1} - \theta_t :\approx -\tilde{\alpha}_t \underbrace{(H_t^r)^{-1}}_{(R)} \underbrace{[H_*^f (H_*^r)^{-1}}_{(S)} \underbrace{[-\nabla \mathcal{L}^f(\theta_t; \varepsilon_t)]]}_{(F)}. \tag{4}$$

We defer the proof to Appendix A.3. The unlearning updates in (4) incorporated second-order Hessian concerning remaining to guide the optimization direction. Specifically, the large curvature direction of $H_t^r$ corresponds with the weights that encapsulate remaining knowledge, while the small curvature direction encourages model updates for effective unlearning. Furthermore, the update direction in (4) primarily follows the weighted gradient ascent (F) modulated by weight saliency (S). Such unlearning iterative updates in remain-preserving manifold focus on diminishing the distributional discrepancy with exact MU concerning forgetting output.

**Challenges in Hessian Approximation.** To exploit the benefits of unlearning updates within the remain-preserving manifold, the key point is $(H_t^r)^{-1}$. However, calculating the Hessian and its reverse for large-scale models is computationally demanding [47]. Consequently, many methods have been developed to estimate the Hessian, such as Fisher information [34], Fisher diagonals [48], and Kronecker factored Laplace approximation [49]. Regarding unlearning, Selective Amnesia (**SA**) [35] employs the initial model's remaining Fisher diagonals as the second-order Hessian constraint on parameter updates. However, the fixed Hessian in SA leads to progressively increasing estimation biases, exacerbated by cumulative errors in Taylor expansion, which harms the retained performance during unlearning. To address this issue, the subsequent **Sec. 4** introduces an fast-slow weight update method that implicitly approximates the direction adjusted by the up-to-date Hessian.

## 4    Proposed Method

**Implicit Online Hessian Approximation (R-on).** Given that computing the inverse of even well-approximated Hessian demands substantial computational resources, it is more practical to estimate the unlearning direction post-Hessian inversion modulation directly. Inspired by recent insights into the connection between Meta-Continual Learning and Hessian approximation [50], we propose a fast-slow weight [36, 37] method for implicitly approximating the desired updates. The optimization problem for fast weight updates is formulated as follows:

$$\min_{\theta_t^f} \mathcal{L}^r(\theta_t^f) \quad \text{s.t.} \quad \theta_t^f = \theta_t - \beta_t \nabla \mathcal{L}^u(\theta_t), \tag{5}$$

where $\mathcal{L}^u$ represents an arbitrary forgetting loss and $\beta_t$ is its learning rate. The iterative process is depicted in **Fig. 1**. A step of forgetting is taken at the current model, resulting in $\theta_t^f$. Several gradient descent updates on the remaining set follow to obtain the minimum point $\theta_t^r$. This fine-tuning ensures that the updated model adheres to the remain-preserving manifold. The slow weight updates leverage the underlying connection between $\theta_t$ and $\theta_t^r$, as stated in **Prop. 3**.

**Proposition 3.** *For implicit online Hessian approximation in (5), suppose $\beta_t, \delta_t$ is small, $\beta_t < \sqrt{\delta_t / |\nabla \mathcal{L}^r(\theta_t) - [\nabla \mathcal{L}^r(\theta_t)]^2|}$, $\mathcal{L}^r$ is $\mu$-smooth, i.e., $\|\nabla \mathcal{L}^r(\theta) - \nabla \mathcal{L}^r(\theta')\|_2 \le \mu \|\theta - \theta'\|_2$, and there exist an $\zeta_t$-neighborhood $\mathcal{N}(\theta_t^r, \zeta_t)$ of the optimal model parameter $\theta_t^r = \arg\min_{\theta_t^f} \mathcal{L}^r(\theta_t^f)$, which includes $\theta_t$ and $\theta_t^f$. Then, the iterative update term approximately is,*

$$\theta_t - \theta_t^r :\approx \beta_t^2 \left[ \nabla^2 \mathcal{L}^r(\theta_t) \right]^{-1} \nabla \mathcal{L}^u(\theta_t) = \beta_t^2 (H_t^r)^{-1} \nabla \mathcal{L}^u(\theta_t). \tag{6}$$

The proof is in Appendix A.4. **Prop. 3** indicates that the model $\theta_t^r$, obtained after fine-tuning using the process described in (5), is approximately equivalent to updating the current model $\theta_t$ by one step in the Hessian-adjusted unlearning direction. We use this direction to update the outer loop.

**Comparison with the joint loss (R).** We investigate the differences in the updates between our optimization in (5) (**R-on**) and the joint optimization of forgetting and remaining losses (**R**) [23, 25]. We take the checkpoint after the first step of fine-tuning the remaining set as an example and ignore the step size.

$$\mathcal{L}^R(\theta_t) = \mathcal{L}^u(\theta_t) + \mathcal{L}^r(\theta_t), \quad \Delta^R = \nabla \mathcal{L}^u(\theta_t) + \nabla \mathcal{L}^r(\theta_t), \tag{7}$$

$$\Delta^{R\text{-on}} \approx \nabla \mathcal{L}^u(\theta_t) + \nabla \mathcal{L}^r(\theta_t) + \nabla^2 \mathcal{L}^r(\theta_t)(\theta_t^f - \theta_t) = (I - H_t^r)\nabla \mathcal{L}^u(\theta_t) + \nabla \mathcal{L}^r(\theta_t), \tag{8}$$

Comparison of the updates in (7) and (8) reveals that the remaining gradient is the same. Our forgetting update in fast weight is adjusted by an additional term $-H_t^r$, which is absent in joint optimization. This modification weakens the directions that significantly impact the remain, thereby mitigating the damage of forgetting loss on the retained performance. Furthermore, certain methods [24] suggest two-stage unlearning that first impairs the model and then repairs it, actually paralleling a single fast-slow weight update of our method.

**Sample-wise Adaptive Coefficient for Gradient Ascent (F).** Despite a variety of forgetting losses introduced in previous literature, we stick to our theoretical result in (4) and adopt the weighted forgetting loss. Due to intractable challenges in solving the inverse problem associated with the $\arg\min$ condition for the gradient ascent coefficients, we explore the properties of these coefficients and propose a heuristic estimation. $\varepsilon_{t,i}$ satisfies: ① For the pretrained model, $\varepsilon_{0,i} = 1$. ② For the retrained model, $\varepsilon_{T^*,i} = 1$, where $T^*$ is the optimal step to obtain RT. ③ Assuming homogeneity of samples, if $\ell(\theta; z_i^f) > \ell(\theta; z_j^f)$, then $\varepsilon_{t,i} < \varepsilon_{t,j}$ [51]. Moreover, considering the continuity in the model parameter space implies the continuity of $\varepsilon_{t,i}$ in the function space. Thus, our heuristic estimation consists of decreasing numerical values across steps and sample-wise adaptation based on loss magnitude,

$$\tilde{\varepsilon}_{t,i} = (1 - \frac{t}{T}) \frac{1/[\ell(\theta_t; z_i^f)]_{\text{detach}}^\lambda}{\sum_{z_j^f \in \mathcal{D}^f} 1/[\ell(\theta_t; z_j^f)]_{\text{detach}}^\lambda} \times N^f, \ 1 \le i \le N^f, \tag{9}$$

where $T$ represents the outer loop iteration, $\lambda$ is the temperature scalar to control the smoothness of coefficients, and $[\cdot]_{\text{detach}}$ denotes the operation to detach a tensor from the computational graph, which means $\ell(\theta_t; z_i^f)$ in (9) only serves for weighting coefficients and does not contribute to the gradient. $\tilde{\varepsilon}_{t,i}$ is employed to modulate the gradient ascent loss for forgetting samples, thereby preventing model explosion and reducing the contribution to updates from samples whose influence has already been ablated, while prioritizing those whose losses remain minimal and are not yet adequately forgotten. However, relying solely on empirical loss as an evaluation metric for sample contribution is limited and potentially biased. We believe that enhanced designs for coefficient estimation in future research could yield more accurate unlearning results.

**Forget-Remain Balanced Weight Saliency (S).** Recall in the theoretical framework of (4), the steepest descent leverages $H_*^f (H_*^r)^{-1}$ as weight saliency to modify the forgetting gradient. However, computing both Hessians at RT is impractical, necessitating an estimation of weight saliency for both forgetting and remaining. Previous work like SalUn [26] considers only parameters that significantly affect the forgetting set, whereas these parameters might also critically impact the retained performance. To address this and align with our theoretical insights, we adopt techniques from SSD [33] to approximate $H_*^f (H_*^r)^{-1}$ using the diagonal of the initial model's Fisher information matrix. Through a hard thresholding operation, we obtain the weight saliency map:

$$\mathbf{m} = \mathbf{I}\left[F_{\text{diag}}^f (F_{\text{diag}}^r)^{-1} \ge \gamma\right], \text{ where } F_{\text{diag}}^f = [\nabla \mathcal{L}^f(\theta_0)]^2, F_{\text{diag}}^r = [\nabla \mathcal{L}^r(\theta_0)]^2. \tag{10}$$

$\mathbf{I}[\cdot]$ is an element-wise indicator function, and $\gamma > 0$ is a hard threshold used to control the Forget-Remain balance in selecting parameters. Guided by (4), we apply the weight saliency map exclusively to forgetting, in contrast to SalUn [26] which applies to both forgetting and remaining. The saliency map can enhance the unlearning process by directing updates to focus on the parameters that are crucial for erasing specific samples or concepts. More advanced weight saliency estimation is expected to improve outcomes in future work.

**Integrated Fast-slow Weight Update.** By integrating the three designs into a single update scheme towards the **S**aliency **F**orgetting in the **R**emain-preserving manifold **on**line, we develop our **SFR-on** method. Specifically, in the inner loop for fast weights, we use adaptive coefficients in (9) to weight the forgetting gradient ascent with the weight saliency map from (10) to serve as the unlearning update in (5). Slow weights in outer loops update by linearly interpolating the fine-tuned $\theta_t^r$ and $\theta_t$ in weight space, achieving an estimated steepest descent for approximate MU under the remaining output constraint in (4). Then, we have the overall fast-slow weight update rule:

$$\text{Inner Loop}: \min_{\theta_t^f} \mathcal{L}^r(\theta_t^r) \quad \text{s.t. } \theta_t^f = \theta_t - \beta_t[\mathbf{m} \odot (-\nabla \mathcal{L}^f(\theta_t; \tilde{\varepsilon}_t))], \tag{11}$$

$$\text{Outer Loop}: \theta_{t+1} = \theta_t - \alpha_t(\theta_t - \theta_t^r) \approx \theta_t - \alpha_t \beta_t^2 (H_t^r)^{-1}[\mathbf{m} \odot (-\nabla \mathcal{L}^f(\theta_t; \tilde{\varepsilon}_t))], \tag{12}$$

where $\alpha_t$ represents the slow weights learning rate. Worth mentioning that our SFR-on does not require adaptation to specific application tasks, nor does it necessitate modifications to the task's inherent loss. Consequently, our approach can be seamlessly applied to various CV unlearning scenarios by simply substituting the loss in the weighted gradient ascent with either CE loss for image classification or MSE loss for image generation. Considering that calculating fine-tuning or the Fisher diagonal on the complete remaining dataset for large-scale image generation tasks, we randomly select an equivalent number of samples from the remaining as in the forgetting set for computation [35, 26]. We find that our method remains effective under such an inadequate setup, as detailed in **Sec. 5**. The complete algorithm is placed in Appendix C.

Table 2: Performance summary of MU methods for image classification (including RT, six baselines, our proposed SFR-on, and ablations on our designed components), assessing unlearning 10% random subset of CIFAR-10 using ResNet-18 and TinyImageNet using Swin-T. All results are presented as mean and standard deviation across 10 independent trials. Performance discrepancies from RT are indicated with (●), highlighting that more effective unlearning is reflected by performance closer to RT. The '*Averaging Disparity*' (Avg.D) metric is calculated by the average of the gaps measured in accuracy-related metrics, including FA, RA, TA, and MIA. $D_{KL}$ denotes the KL divergence to RT. RTE is recorded in minutes.

| Methods | CIFAR-10 Random Subset Forgetting (10%) | | | | | | | TinyImageNet Random Subset Forgetting (10%) | | | | | | |
|---|---|---|---|---|---|---|---|---|---|---|---|---|---|---|
| | FA | RA | TA | MIA | Avg.D↓ | $D_{KL}$↓ | RTE | FA | RA | TA | MIA | Avg.D↓ | $D_{KL}$↓ | RTE |
| RT | $95.62_{\pm0.25}$ (0.00) | $100.00_{\pm0.00}$ (0.00) | $95.34_{\pm0.08}$ (0.00) | $74.84_{\pm0.00}$ (0.00) | 0.00 | 0.10 | 73.37 | $85.29_{\pm0.09}$ (0.00) | $99.55_{\pm0.03}$ (0.00) | $85.49_{\pm0.15}$ (0.00) | $69.30_{\pm0.20}$ (0.00) | 0.00 | 0.18 | 42.01 |
| FT | $99.90_{\pm0.05}$ (4.28) | $99.99_{\pm0.00}$ (0.01) | $94.94_{\pm0.15}$ (0.39) | $88.25_{\pm0.01}$ (13.42) | 4.52 | 0.26 | 3.83 | $96.45_{\pm0.13}$ (11.16) | $98.29_{\pm0.08}$ (1.26) | $82.46_{\pm0.16}$ (3.03) | $90.00_{\pm0.22}$ (20.70) | 9.04 | 0.60 | 4.38 |
| GA | $93.91_{\pm1.67}$ (1.71) | $93.76_{\pm1.89}$ (6.24) | $87.00_{\pm1.64}$ (8.34) | $77.19_{\pm0.01}$ (2.35) | 4.66 | 0.36 | 0.79 | $83.28_{\pm4.18}$ (2.01) | $84.55_{\pm4.63}$ (15.00) | $70.98_{\pm3.61}$ (14.51) | $73.86_{\pm3.31}$ (4.56) | 9.02 | 1.09 | 4.13 |
| RL | $95.99_{\pm0.24}$ (0.38) | $99.98_{\pm0.01}$ (0.02) | $93.85_{\pm0.11}$ (1.48) | $31.44_{\pm0.01}$ (43.40) | 11.32 | 0.34 | 4.56 | $93.35_{\pm0.31}$ (8.06) | $98.15_{\pm0.14}$ (1.40) | $82.98_{\pm0.22}$ (2.51) | $45.29_{\pm1.04}$ (24.00) | 9.00 | 0.47 | 4.79 |
| SalUn | $100.00_{\pm0.01}$ (4.38) | $99.99_{\pm0.01}$ (0.01) | $94.89_{\pm0.09}$ (0.45) | $67.54_{\pm0.00}$ (7.29) | 3.03 | 0.27 | 4.58 | $95.78_{\pm0.25}$ (10.49) | $98.60_{\pm0.06}$ (0.95) | $83.63_{\pm0.22}$ (1.87) | $51.18_{\pm1.92}$ (18.12) | 7.86 | 0.48 | 4.88 |
| BT | $98.88_{\pm0.00}$ (3.26) | $99.99_{\pm0.00}$ (0.01) | $94.63_{\pm0.06}$ (0.71) | $61.77_{\pm0.00}$ (13.07) | 4.26 | 0.24 | 5.56 | $93.22_{\pm0.30}$ (7.93) | $97.82_{\pm0.14}$ (1.73) | $83.04_{\pm0.22}$ (2.45) | $47.53_{\pm0.71}$ (21.77) | 8.47 | 0.47 | 6.79 |
| SCRUB | $99.44_{\pm0.31}$ (3.82) | $99.88_{\pm0.08}$ (0.12) | $94.13_{\pm0.35}$ (1.20) | $87.43_{\pm0.00}$ (12.59) | 4.43 | 0.25 | 2.56 | $97.23_{\pm0.05}$ (11.94) | $98.10_{\pm0.34}$ (1.45) | $82.74_{\pm0.21}$ (2.75) | $81.32_{\pm0.47}$ (12.02) | 7.04 | 0.62 | 5.49 |
| S F R on | | | | | | | | | | | | | | |
| ✓ | $96.38_{\pm0.35}$ (0.76) | $99.66_{\pm0.21}$ (0.34) | $91.96_{\pm0.31}$ (3.38) | $83.16_{\pm0.47}$ (8.32) | 3.20 | 0.32 | 3.13 | $89.90_{\pm0.39}$ (4.61) | $94.05_{\pm0.19}$ (5.50) | $77.98_{\pm0.72}$ (7.51) | $78.16_{\pm0.93}$ (8.86) | 6.62 | 0.73 | 6.10 |
| ✓ ✓ | $96.84_{\pm0.50}$ (1.22) | $99.92_{\pm0.21}$ (0.08) | $94.18_{\pm0.28}$ (1.16) | $80.38_{\pm0.25}$ (5.54) | 2.00 | 0.23 | 2.12 | $93.42_{\pm0.16}$ (8.13) | $98.92_{\pm0.04}$ (0.63) | $83.45_{\pm0.21}$ (2.04) | $81.84_{\pm0.77}$ (12.54) | 5.83 | 0.73 | 4.02 |
| ✓ ✓ ✓ | $96.16_{\pm0.72}$ (0.54) | $99.98_{\pm0.20}$ (0.02) | $94.24_{\pm0.30}$ (1.10) | $70.64_{\pm0.26}$ (4.20) | 1.47 | 0.20 | 2.12 | $95.51_{\pm0.29}$ (10.22) | $98.79_{\pm0.04}$ (0.76) | $83.11_{\pm0.13}$ (2.38) | $64.00_{\pm0.87}$ (5.30) | 4.67 | 0.45 | 4.02 |
| ✓ ✓ ✓ ✓ | $96.58_{\pm0.77}$ (0.96) | $99.88_{\pm0.16}$ (0.12) | $94.19_{\pm0.33}$ (1.15) | $72.26_{\pm0.01}$ (2.58) | 1.20 | 0.15 | 2.80 | $97.02_{\pm0.16}$ (11.73) | $99.18_{\pm0.05}$ (0.37) | $84.00_{\pm0.18}$ (1.49) | $71.09_{\pm0.76}$ (1.79) | 3.85 | 0.44 | 4.21 |

# 5 Experiments

**Datasets, Models, and Settings.** In image classification, we primarily focus on the **random subset unlearning** task. Evaluations are conducted using ResNet-18 [52] on CIFAR10 [53] and Swin-T [54] on TinyImageNet [55], with additional tests on random subset and class-wise forgetting tasks involving CIFAR100 [53] and SVHN [56], detailed in Appendix F.2. In image generation, our main interest lies in **class-wise forgetting** tasks. Following [35, 26], we unlearn conditional DDPM [40] with the UNet architecture [57] on CIFAR10. Moreover, for the first time, we explore the latent diffusion model [42] equipped with Diffusion Transformer (DiT) [58] on ImageNet [59], which demonstrates superior scalability in learning large-scale data generation tasks. Finally, we perform **concept forgetting** tasks using the open-source Stable Diffusion (SD) V1.4 [42] to inhibit the generation of NSFW content, specifically by targeting the prevention of nude images. Further details on unlearning setups and training are available in Appendix E.

**Baselines and Evaluation.** We regard RT as an oracle of approximate MU and compare our proposal with eight MU methods, including six gradient-based MU approaches outlined in **Sec. 3**: **FT** [19], **GA** [21], **BT** [23], **RL** [20], **SalUn** [26], and **SA** [35]. We also consider **SCRUB** [25], an enhanced variant of BT, for image classification, and **ESD** [11] for removing concepts in SD. For the evaluation of random subset unlearning tasks, we measure the output KL divergence $D_{KL}$ between the unlearned and retrained models, which is the direct target of approximate MU. Besides, we assess the accuracy on the forgetting set (**FA**) for unlearning efficacy, and the accuracy on the remaining (**RA**) and test (**TA**) sets for preserved generalization ability. We also consider the success rate of membership inference attack (**MIA**) [60, 38] on the forgetting set as a privacy metric. Note that the smaller the disparity in these metrics against RT, the more effective the unlearning. Run-time efficiency (**RTE**) is also reported. For class-wise forgetting tasks in image generation, we evaluate the accuracy of the unlearned model's generated images on forgetting classes (**FA**) by a pre-trained classifier. The Fréchet Inception Distance (**FID**) [61] metric assesses the retained generative capability for remaining classes. For ablating the 'nudity' concept in SD, we employ the NudeNet [62] detector to identify and count nude body parts in generated NSFW images. For further introduction to the baselines and detailed evaluation metrics, please refer to Appendix E.1 and D.

To assess the unlearning effectiveness and efficiency of **SFR-on**, we perform comprehensive experiments and conduct ablation studies to address the following **four key questions**:

**Q1: How does SFR-on perform on unlearning in image classification?** We first evaluate the performance of our method, SFR-on, against existing gradient-based MU methods on the image classification random subset unlearning task. In this scenario, the forgetting set, remaining, and test sets all originate from the same distribution. Consequently, even if the model undergoes unlearning on the random subset, it may still generalize to these samples. To avoid potential biases from only using FA as an unlearning metric, we incorporate MIA to assess the privacy retention of the forgetting set, enhancing the robustness of assessments. As detailed in **Tab. 2**, for forgetting 10% random subset on CIFAR-10 and TinyImageNet, SFR-on not only most closely aligns with RT in the averaging metric disparity but also exhibits the smallest output KL divergences *w.r.t.* RT. This performance underscores our effectiveness and efficiency in achieving the objective of approximate MU. The results of the increased 50% random subset unlearning task are included in Appendix F.2.

Table 3: Class-wise forgetting performance on CIFAR10 with DDPM and ImageNet with DiT. The best unlearning performance for each forgetting class is highlighted in **bold** for FA and FID.

| Methods | CIFAR-10 Class-wise Forgetting | | | | | | | | | | | Steps | ImageNet Class-wise Forgetting | | | | | | | | | | Steps |
|---|---|---|---|---|---|---|---|---|---|---|---|---|---|---|---|---|---|---|---|---|---|---|---|
| | Automobile | | Cat | | Dog | | Horse | | Truck | | | | Cacatua galerita | | Golden retriever | | White wolf | | Arctic fox | | Otter | | |
| | FA↓ | FID↓ | FA↓ | FID↓ | FA↓ | FID↓ | FA↓ | FID↓ | FA↓ | FID↓ | | | FA↓ | FID↓ | FA↓ | FID↓ | FA↓ | FID↓ | FA↓ | FID↓ | FA↓ | FID↓ | |
| SA | 0.00 | 23.56 | 14.20 | 21.34 | 8.60 | 21.19 | 0.00 | 21.13 | 0.00 | 29.04 | 10000 | | 0.00 | 348.75 | 0.00 | 298.97 | 0.00 | 45.89 | 0.00 | 393.91 | 29.8 | 321.21 | 10000 |
| SalUn | 0.20 | 21.23 | **1.40** | 20.29 | 0.00 | 20.18 | 0.60 | 20.70 | 0.80 | **20.45** | 1000 | | 91.21 | 18.47 | 46.09 | 25.28 | 0.00 | **15.16** | 45.90 | 408.07 | 87.50 | 19.69 | 10000 |
| SFR-on | 0.00 | **20.70** | 7.40 | **18.44** | 0.20 | **18.89** | 0.00 | **19.93** | 0.00 | 20.61 | 50 | | 0.00 | **13.59** | 0.00 | **17.76** | 0.00 | 23.28 | 0.00 | **16.12** | 0.00 | **16.43** | 500 |

Figure 2: Image generations for class-wise forgetting tasks on CIFAR-10 using DDPM by baselines and our proposed SFR-on along with ablation variants. The forgetting class is 'cat', 'I' refers to the generated *image sample* from this class, and 'C' denotes the remaining *class name*. More results can be found in Appendix F.7.

**Q2: How does SFR-on perform on class-forgetting in image generation?** Another recent focus in MU involves the targeted removal of specific knowledge from image generation models, which are currently categorized into conditional and latent diffusion models. For the former, we investigated class-wise forgetting on CIFAR-10 using DDPM. As illustrated in **Fig. 2**, RT continues to generate high-quality images devoid of the semantic content of 'cat.' Contrarily, previous approaches like SA resulted in random noise for the forgetting class, whereas SalUn created images belonging to alternate classes, both differ from RT output. Our SFR-on (depicted in the last row of **Fig. 2**) effectively removes the 'cat' class by yielding high-quality pictures without discernible semantics, and maintains the high fidelity of images across non-forgetting classes, most matching RT. Furthermore, we extend the image generation unlearning task to DiT, a recently proposed model promising for realistic image generation. Due to computational resource constraints, RT for DiT is unfeasible. Instead, we simulate RT by substituting the DiT output of the forgetting class with initial random latent embeddings. These embeddings are then processed by the pre-trained autoencoder to reconstruct the corresponding images, referred to as RT[†] in **Fig. 3**. Previous methods, such as SA and SalUn, fail to completely reduce the forgetting class to noise and compromise the fidelity of non-forgetting class images. In contrast, our SFR-on successfully achieves results comparable to RT[†], effectively forgetting the target class without degrading the general generative capability. Details on the accuracy of forgetting class samples by various unlearned DiT and FID of remaining classes are presented in **Tab. 3**.

**Q3: What is the impact of each component of SFR-on on forgetting performance?** To investigate the efficacy of each component we developed for the decomposition of approximate MU, we perform ablation studies. We build on the joint training of GA and FT as our baseline (R) and incorporate our proposed implicit online hessian approximation (R-on), adaptive coefficients (F), and forget-retain balanced weight saliency (S). As demonstrated in the last four rows of **Tab. 2**, the addition of our (R-on) allows models to effectively forget while sustaining performance on the remaining set. Both (F) and (S), crafted to approach the steepest descent for approximate MU, enhance the unlearning efficacy. In image generation, as depicted in the last three rows of **Fig. 2**, although the joint training (R) is capable of completely forgetting targeted classes, it results in distorted images for remaining. Replacing (R) with our (R-on) remarkably improves the image fidelity of the remaining classes, but the forgetting class images still show low-quality textures. Further, our (F) and (S) effectively direct the unlearning process towards the approximate MU, ensuring that the performance of the unlearned models closely mirrors that of RT. More ablations on hyperparameters are provided in Appendix F.1.

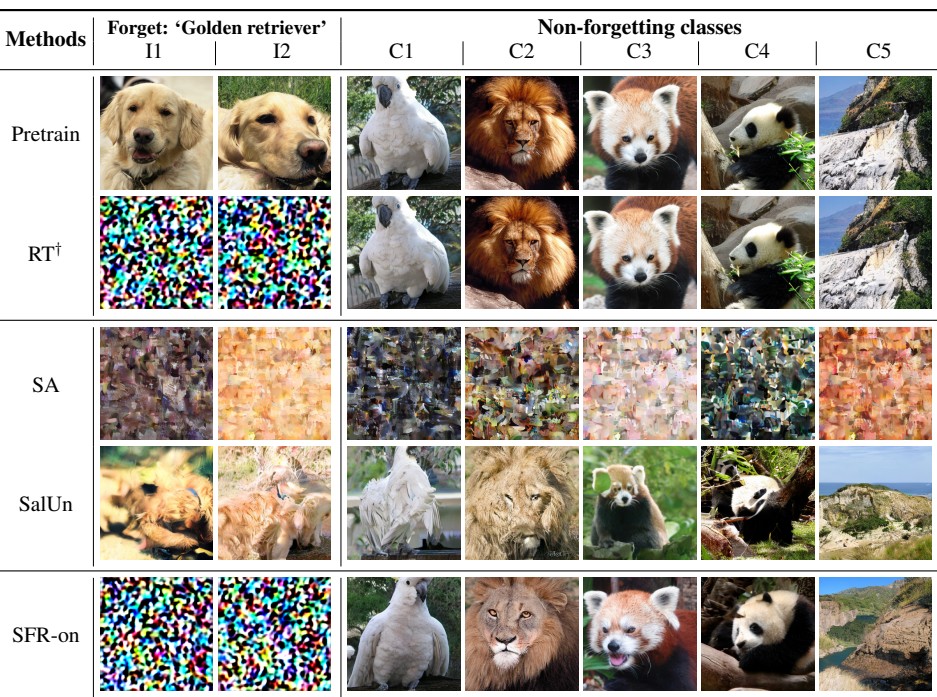

| Methods | Forget: 'Golden retriever' | | Non-forgetting classes | | | | |
|---|---|---|---|---|---|---|---|
| | I1 | I2 | C1 | C2 | C3 | C4 | C5 |
| Pretrain | | | | | | | |
| RT$^\dagger$ | | | | | | | |
| SA | | | | | | | |
| SalUn | | | | | | | |
| SFR-on | | | | | | | |

Figure 3: Class-wise forgetting of 'golden retriever' in image generations of ImageNet with DiT, comparing baselines and our proposed SFR-on. RT$^\dagger$ feeds autoencoder with random latent embeddings for the forgetting class, due to the computational constraints, rather than full RT. 'I' denotes *image samples* from forgetting, and 'C' refers to other remaining *class name*, *e.g.* 'Cacatua galerita' (C1). More results can be found in Appendix F.7.

**Q4: How effective is SFR-on in NSFW content removal for SD?** We finally assess the efficacy of our method in removing the 'nudity' concept from the open-source SD to prevent the generation of NSFW content. Given that SD V1.4 is trained on the LAION dataset [63], purging all images depicting nudity and retraining the model would be prohibitively time-consuming and resource-intensive. In this unlearning task, we designate 'nudity' as the forgetting set and generate a set of clothed individuals as remaining to preserve SD's generalization capability across non-nudity themes. We utilize the inappropriate image prompts (I2P) [12] to query potentially NSFW content from the unlearned SD and employ NudeNet to detect exposed body parts in these images. The results in **Tab. 4** demonstrate that our method significantly prevents the generation of culturally sensitive body parts, such as breasts and genitalia, highlighting our strength to enhance the trustworthiness of machine learning applications.

Table 4: Unlearning performance in erasing 'Nudity' concept from SD by the original SD V1.4, ESD, SalUn, and our SFR-on, measuring the number of total 9406 NSFW images generated from I2P prompts across nudity categories. The prefixes 'F-' and 'M-' denote 'Female-' and 'Male-' respectively.

| Methods | # of exposed body parts ↓ | | | | | | | | | |
|---|---|---|---|---|---|---|---|---|---|---|
| | Buttocks | F-Breast | F-Genitalia | M-Genitalia | M-Breast | Anus | Feet | Armpits | Belly | Total |
| SD V1.4 | 107 | 298 | 56 | 143 | 351 | 4 | 168 | 409 | 263 | 1799 |
| ESD | 35 | 73 | 9 | 51 | 79 | 1 | 121 | 81 | 42 | 492 |
| SalUn | 2 | 2 | 1 | 5 | 3 | 0 | 9 | 21 | 3 | 46 |
| SFR-on | 0 | 0 | 0 | 0 | 0 | 0 | 1 | 2 | 0 | 3 |

## 6 Conclusion

This paper revisits gradient-based approximate MU methods from the perspective of the steepest descent. The descent direction under an Euclidean manifold metric can be divided into three integral components: weighted forgetting gradient ascent, fine-tuning remaining gradient descent, and weight saliency matrix. Our approach advances beyond the Euclidean constraints by embedding the unlearning update within a remain-preserving manifold. This novel strategy incorporates the second-order Hessian of the current model on remaining, safeguarding against detrimental impacts on retained performance. To circumvent the prohibitive computational demands of the Hessian in large-scale models, we introduce an efficient fast-slow weight update method to approximate the Hessian-adjusted direction. Furthermore, our innovative adaptive coefficient for weight forgetting loss and a forget-remain balanced weight saliency map facilitate near-retraining unlearning. Our method can be applied to popular CV unlearning tasks with empirically verified unlearning efficacy.

## Acknowledgments and Disclosure of Funding

The authors would like to thank the anonymous reviewers for their insightful comments.

The research leading to these results has received funding from National Key Research Development Project (2023YFF1104202), National Natural Science Foundation of China (62376155, Shanghai Municipal Science and Technology Research Program Major Project (2021SHZDZX0102).

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

# Appendix

## A    Detailed Proof

### A.1    Proof of Equation (1)

*Proof.* We form the following optimization problem for the steepest descent of approximate MU, which is to find the direction $\delta\theta = \theta_{t+1} - \theta_t$ that drives the objective function $F(\theta)$ descent fastest within a $\xi$-neighborhood of the current parameters $\theta_t$, and the $\xi$-neighborhood is rendered by the manifold metric $\rho(\cdot,\cdot)$:

$$\delta\theta = \underset{\delta\theta}{\arg\min}\, F(\theta_t + \delta\theta) \quad \text{s.t.} \quad \rho(\theta_t, \theta_t + \delta\theta) \leq \xi. \tag{A1}$$

We introduce a Lagrange multiplier $\eta \geq 0$ to construct the Lagrangian $\tilde{\mathcal{L}}$ for this optimization problem:

$$\tilde{\mathcal{L}}(\delta\theta, \eta) = F(\theta_t + \delta\theta) + \eta(\rho(\theta_t, \theta_t + \delta\theta) - \xi). \tag{A2}$$

Using the Karush-Kuhn-Tucker (KKT) theorem, we can take the derivative of $\tilde{\mathcal{L}}$ w.r.t. $\delta\theta$ and set it to zero:

$$\nabla_{\delta\theta}\tilde{\mathcal{L}}(\delta\theta, \eta) = \nabla_{\delta\theta}F(\theta_t + \delta\theta) + \eta\nabla_{\delta\theta}\rho(\theta_t, \theta_t + \delta\theta) = 0, \tag{A3}$$

where $\eta$ depends on $\xi$ and $\theta_t$ implicitly. We can rewrite (A3) by variable substitution $\theta_{t+1} = \theta_t + \delta\theta$ and $\eta = 1/\alpha_t(\xi, \theta_t)$.

$$\nabla_{\theta_{t+1}}\tilde{\mathcal{L}}(\theta_{t+1}, \eta) = \nabla_{\theta_{t+1}}F(\theta_{t+1}) + \frac{1}{\alpha_t(\xi, \theta_t)}\nabla_{\theta_{t+1}}\rho(\theta_t, \theta_{t+1}) = 0. \tag{A4}$$

Thus, the original problem is transformed into an unconstrained optimization problem w.r.t. $\theta_{t+1}$, where the neighborhood size is implicitly given by $\alpha_t$:

$$\theta_{t+1} = \underset{\theta_{t+1}}{\arg\min}\, F(\theta_{t+1}) + \frac{1}{\alpha_t(\xi, \theta_t)}\rho(\theta_t, \theta_{t+1}). \tag{A5}$$

$\square$

### A.2    Proof of Proposition 1

The optimization problem of **Prop. 1** is to find the steepest descent direction that minimizes the KL divergence with the retrained output within the vicinity of the current model $\theta_t$:

$$\theta_{t+1} = \underset{\theta_{t+1}}{\arg\min}\, D_{\text{KL}}\left(p_z(\theta_*)||p_z(\theta_{t+1})\right) + \frac{1}{\alpha_t}\rho(\theta_t, \theta_{t+1}) \tag{A6}$$

$$= \underset{\theta_{t+1}}{\arg\min}\, D_{\text{KL}}\left(p_{z^f}(\theta_*)||p_{z^f}(\theta_{t+1})\right)p^f + D_{\text{KL}}\left(p_{z^r}(\theta_*)||p_{z^r}(\theta_{t+1})\right)p^r + \frac{1}{\alpha_t}\rho(\theta_t, \theta_{t+1})$$

**Proposition 1.** *Under the Euclidean manifold metric, $\rho(\theta_t, \theta_{t+1}) = \frac{1}{2}\|\theta_t - \theta_{t+1}\|^2$. Assuming that the current model $\theta_t = \arg\min_\theta \mathcal{L}^r(\theta) + \mathcal{L}^f(\theta; \varepsilon_t)$. Let $H_*^f = \nabla^2\mathcal{L}^f(\theta_*; \mathbf{1})$ and $H_*^r = \nabla^2\mathcal{L}^r(\theta_*)$ denote the Hessian matrix of the retrained model on the forgetting set and the remaining set, respectively. Then, the direction of the steepest gradient descent that minimizes the KL divergence between the output of the current model and the retrained model is approximately:*

$$\theta_{t+1} - \theta_t :\approx -\alpha_t[H_*^f(H_*^r)^{-1}[-\nabla\mathcal{L}^f(\theta_t; \varepsilon_t)]p^f + \nabla\mathcal{L}^r(\theta_t)p^r] \tag{A7}$$

*Proof.* We can decompose the original optimization problem into three parts: $F(\theta_{t+1})$, $R(\theta_{t+1})$, and $C(\theta_{t+1})$.

$$
\begin{aligned}
\theta_{t+1} &= \underset{\theta_{t+1}}{\arg\min}\, D_{\mathrm{KL}}\left(p_z(\theta_*)||p_z(\theta_{t+1}))\right) + \frac{1}{2\alpha_t}\|\theta_t - \theta_{t+1}\|^2 \\
&= \underset{\theta_{t+1}}{\arg\min} \int p(z;\theta_*)\left[\log p(z;\theta_*) - \log p(z;\theta_{t+1})\right]\mathrm{d}z + \frac{1}{2\alpha_t}\|\theta_t - \theta_{t+1}\|^2 \\
&= \underset{\theta_{t+1}}{\arg\min} \int p(z^f;\theta_*)\left[\log p(z^f;\theta_*) - \log p(z^f;\theta_{t+1})\right]\mathrm{d}z^f\, p(\mathcal{D}^f|\mathcal{D}) \\
&\quad + \int p(z^r;\theta_*)\left[\log p(z^r;\theta_*) - \log p(z^r;\theta_{t+1})\right]\mathrm{d}z^r\, p(\mathcal{D}^r|\mathcal{D}) \\
&\quad + \frac{1}{2\alpha_t}\|\theta_t - \theta_{t+1}\|^2 \\
&= \underset{\theta_{t+1}}{\arg\min}\, \mathbb{E}_{p(z^f;\theta_*)}\left[\log p(z^f;\theta_*) - \log p(z^f;\theta_{t+1})\right] p(\mathcal{D}^f|\mathcal{D}) \\
&\quad + \mathbb{E}_{p(z^r;\theta_*)}\left[\log p(z^r;\theta_*) - \log p(z^r;\theta_{t+1})\right] p(\mathcal{D}^r|\mathcal{D}) \\
&\quad + \frac{1}{2\alpha_t}\|\theta_t - \theta_{t+1}\|^2 \\
&= \underset{\theta_{t+1}}{\arg\min} \underbrace{D_{\mathrm{KL}}\left(p_{z^f}(\theta_*)||p_{z^f}(\theta_{t+1}))\right)}_{(F(\theta_{t+1}))} p^f + \underbrace{D_{\mathrm{KL}}\left(p_{z^r}(\theta_*)||p_{z^r}(\theta_{t+1}))\right)}_{(R(\theta_{t+1}))} p^r \\
&\quad + \frac{1}{2\alpha_t}\underbrace{\|\theta_t - \theta_{t+1}\|^2}_{(C(\theta_{t+1}))}
\end{aligned}
\tag{A8}
$$

(**Part I**) First, we solve the forgetting part $F(\theta_{t+1})$ by taking the first-order approximation at $\theta_t$:

$$
F(\theta_{t+1}) = F(\theta_t) + \nabla F(\theta_t)^\top (\theta_{t+1} - \theta_t).
\tag{A9}
$$

We denote $\nabla F(\theta_t) = -\mathbb{E}_{p(z^f;\theta_*)}\left[\nabla \log p(z^f;\theta_t)\right] = G(\theta_t)$, and then expand $G(\theta_t)$ at $\theta_*$:

$$
\begin{aligned}
G(\theta_t) &= G(\theta_*) + \nabla G(\theta_*)(\theta_t - \theta_*) \\
&= -\mathbb{E}_{p(z^f;\theta_*)}\left[\nabla \log p(z^f;\theta_*)\right] - \mathbb{E}_{p(z^f;\theta_*)}\left[\nabla^2 \log p(z^f;\theta_*)\right]\Delta_t \\
&= 0 + H_*^f \Delta_t,
\end{aligned}
\tag{A10}
$$

where $H_*^f = -\mathbb{E}_{p(z^f;\theta_*)}\left[\nabla^2 \log p(z^f;\theta_*)\right]$ is the Hessian $w.r.t.$ the retrained model $\theta_*$ at forgetting set, and $\Delta_t = \theta_t - \theta_*$ is the difference between the current model $\theta_t$ and the retrained model $\theta_*$. We cannot directly obtain the parameter difference $\Delta_t$ and need to estimate it. Recalling the assumption that $\theta_t = \arg\min_\theta \mathcal{L}^r(\theta) + \mathcal{L}^f(\theta;\varepsilon_t)$ and $\theta_* = \arg\min_\theta \mathcal{L}^r(\theta)$. We can utilize the optimality of $\theta_t$ on the weighted function to take the derivative $w.r.t.$ $\theta_t$ and set it to zero:

$$
\begin{aligned}
0 &= \nabla\mathcal{L}^r(\theta_t) + \nabla\mathcal{L}^f(\theta_t;\varepsilon_t) \\
&= \left[\nabla\mathcal{L}^r(\theta_*) + \nabla^2 L^r(\theta_*)\Delta_t + o(\Delta_t)\right] + \nabla\mathcal{L}^f(\theta_t;\varepsilon_t) \\
&\approx 0 + \nabla^2\mathcal{L}^r(\theta_*)\Delta_t + \nabla\mathcal{L}^f(\theta_t;\varepsilon_t).
\end{aligned}
\tag{A11}
$$

Since $\theta_*$ minimizes $\mathcal{L}^r$, $\nabla\mathcal{L}^r(\theta_*) = 0$. By performing the Taylor expansion and dropping $o(\Delta_t)$ terms, we have

$$
\Rightarrow \Delta_t \approx -\left[\nabla^2\mathcal{L}^r(\theta_*)\right]^{-1}\nabla\mathcal{L}^f(\theta_t;\varepsilon_t) = -(H_*^r)^{-1}\nabla\mathcal{L}^f(\theta_t;\varepsilon_t).
\tag{A12}
$$

By plugging (A12) into (A10), we can get

$$
\nabla F(\theta_t) = G(\theta_t) \approx -H_*^f(H_*^r)^{-1}\nabla\mathcal{L}^f(\theta_t;\varepsilon_t).
\tag{A13}
$$

Bringing (A13) into (A9), we can get

$$
F(\theta_{t+1}) \approx F(\theta_t) - \left[H_*^f(H_*^r)^{-1}\nabla\mathcal{L}^f(\theta_t;\varepsilon_t)\right]^\top (\theta_{t+1} - \theta_t).
\tag{A14}
$$

(**Part II**) Next, we solve the remaining part $R(\theta_{t+1})$ with similar pipeline in solving $F(\theta_{t+1})$:

$$
\begin{aligned}
R(\theta_{t+1}) &= R(\theta_t) + \nabla R(\theta_t)^\top (\theta_{t+1} - \theta_t) \\
&= R(\theta_t) - \mathbb{E}_{p(z^r;\theta_*)} \left[ \nabla \log p(z^r; \theta_t) \right]^\top (\theta_{t+1} - \theta_t) \\
&= R(\theta_t) + \left[ \nabla \mathcal{L}^r(\theta_t) \right]^\top (\theta_{t+1} - \theta_t).
\end{aligned}
\tag{A15}
$$

(**Part III**) Finally, we derive the constraint $C(\theta_{t+1})$ as follows,

$$
\begin{aligned}
C(\theta_{t+1}) =& C(\theta_t) + \nabla C(\theta_t)^\top (\theta_{t+1} - \theta_t) \\
=& C(\theta_t) + 2(\theta_{t+1} - \theta_t)^\top (\theta_{t+1} - \theta_t).
\end{aligned}
\tag{A16}
$$

Substituting (A14), (A15), and (A16) to each part in (A8), and take the derivative $w.r.t.$ $\theta_{t+1}$ of the minimization problem to derive the optimal solution, we have

$$
\begin{aligned}
0 &= \nabla F(\theta_{t+1}) p^f + \nabla R(\theta_{t+1}) p^r + \frac{1}{2\alpha_t} \nabla C(\theta_{t+1}) \\
&\approx H_*^f (H_*^r)^{-1} \left[ -\nabla \mathcal{L}^f(\theta_t; \varepsilon_t) \right] p^f + \nabla \mathcal{L}^r(\theta_t) p^r + \frac{1}{\alpha_t} (\theta_{t+1} - \theta_t).
\end{aligned}
\tag{A17}
$$

We thus conclude that

$$
\Rightarrow \theta_{t+1} - \theta_t \approx -\alpha_t \left[ H_*^f (H_*^r)^{-1} [-\nabla \mathcal{L}^f(\theta_t; \varepsilon_t)] p^f + \nabla \mathcal{L}^r(\theta_t) p^r \right].
\tag{A18}
$$

$\square$

### A.3 Proof of Proposition 2

**Proposition 2.** *Using the model output KL divergence on the remaining set as the manifold metric, $\rho(\theta_t, \theta_{t+1}) = D_{KL}(p_{z^r}(\theta_t) || p_{z^r}(\theta_{t+1})))$. Assuming that the current model $\theta_t = \arg\min_\theta \mathcal{L}^r(\theta) + \mathcal{L}^f(\theta; \varepsilon_t)$. Let $\tilde{\alpha}_t = \alpha_t p^f / (\alpha_t p^r + 1)$, and $H_t^r = \nabla^2 \mathcal{L}^r(\theta_t)$ represents the Hessian matrix of the current model on the remaining set, then the direction of the steepest gradient descent that minimizes the KL divergence between the output of the current model and the retrained model is approximately:*

$$
\theta_{t+1} - \theta_t :\approx -\tilde{\alpha}_t (H_t^r)^{-1} \left[ H_*^f (H_*^r)^{-1} [-\nabla \mathcal{L}^f(\theta_t; \varepsilon_t)] \right]
\tag{A19}
$$

*Proof.* Now, the steepest descent optimization problem for approximate MU is as follows:

$$
\begin{aligned}
\theta_{t+1} =& \arg\min_{\theta_{t+1}} D_{\mathrm{KL}}(p_z(\theta_*) || p_z(\theta_{t+1}))) + \frac{1}{\alpha_t} D_{\mathrm{KL}}(p_{z^r}(\theta_t) || p_{z^r}(\theta_{t+1}))) \\
=& \arg\min_{\theta_{t+1}} \underbrace{D_{\mathrm{KL}}(p_{z^f}(\theta_*) || p_{z^f}(\theta_{t+1})))}_{(F(\theta_{t+1}))} p^f + \underbrace{D_{\mathrm{KL}}(p_{z^r}(\theta_*) || p_{z^r}(\theta_{t+1})))}_{(R(\theta_{t+1}))} p^r \\
& + \frac{1}{\alpha_t} \underbrace{D_{\mathrm{KL}}(p_{z^r}(\theta_t) || p_{z^r}(\theta_{t+1})))}_{(C(\theta_{t+1}))}
\end{aligned}
\tag{A20}
$$

The result of forgetting part $F(\theta_{t+1})$ is the same as that in the Euclidean distance metric. And the remaining part $R(\theta_{t+1})$ and the constraint $C(\theta_{t+1})$ vary due to the output KL divergence metric $D_{\mathrm{KL}}^r$. Note that $\nabla \mathcal{L}^r(\theta_{t+1}) \approx \nabla \mathcal{L}^r(\theta_t) \approx \nabla \mathcal{L}^r(\theta_0) \approx 0$. This enables us to take the second-order Taylor expansion at $\theta_t$ for the remaining part and the constraint.

$$
R(\theta_{t+1}) = R(\theta_t) + \nabla R(\theta_t)^\top (\theta_{t+1} - \theta_t) + \frac{1}{2} (\theta_{t+1} - \theta_t)^\top \nabla^2 R(\theta_t) (\theta_{t+1} - \theta_t)
\tag{A21}
$$

$$
\nabla R(\theta_t) = -\mathbb{E}_{p(z^r;\theta_*)} \left[ \nabla \log p(z^r; \theta_t) \right] = \nabla \mathcal{L}^r(\theta_t) \approx 0
\tag{A22}
$$

$$
\nabla^2 R(\theta_t) = -\mathbb{E}_{p(z^r;\theta_*)} \left[ \nabla^2 \log p(z^r; \theta_t) \right] = \nabla^2 \mathcal{L}^r(\theta_t) = H_t^r
\tag{A23}
$$

Substituting (A22) and (A23) into (A21), we can derive the remaining part:

$$
R(\theta_{t+1}) \approx R(\theta_t) + \frac{1}{2} (\theta_{t+1} - \theta_t)^\top H_t^r (\theta_{t+1} - \theta_t)
\tag{A24}
$$

As for the constraint, we have

$$C(\theta_{t+1}) = C(\theta_t) + \nabla C(\theta_t)^\top (\theta_{t+1} - \theta_t) + \frac{1}{2}(\theta_{t+1} - \theta_t)^\top \nabla^2 C(\theta_t)(\theta_{t+1} - \theta_t) \qquad (A25)$$

$$\nabla C(\theta_t) = -\mathbb{E}_{p(z^r;\theta_t)}[\nabla \log p(z^r;\theta_t)] = 0 \qquad (A26)$$

$$\nabla^2 C(\theta_t) = -\mathbb{E}_{p(z^r;\theta_t)}[\nabla^2 \log p(z^r;\theta_t)] = F_t^r \approx H_t^r = \nabla^2 \mathcal{L}^r(\theta_t) \qquad (A27)$$

Substituting (A26) and (A27) into (A25), we get the constraint

$$C(\theta_{t+1}) \approx C(\theta_t) + \frac{1}{2}(\theta_{t+1} - \theta_t)^\top H_t^r(\theta_{t+1} - \theta_t). \qquad (A28)$$

Bringing (A14), (A24), and (A28) into (A20), and take the derivative w.r.t. $\theta_{t+1}$ of the minimization problem to derive the optimal solution, we have

$$0 = \nabla F(\theta_{t+1})p^f + \nabla R(\theta_{t+1})p^r + \frac{1}{\alpha_t}\nabla C(\theta_{t+1}) \qquad (A29)$$

$$\approx H_*^f (H_*^r)^{-1}[-\nabla \mathcal{L}^f(\theta_t;\varepsilon_t)]p^f + H_t^r(\theta_{t+1} - \theta_t)p^r + \frac{1}{\alpha_t}H_t^r(\theta_{t+1} - \theta_t). \qquad (A30)$$

By rearranging the terms, we get

$$\Rightarrow \theta_{t+1} - \theta_t :\approx -\tilde{\alpha}_t(H_t^r)^{-1}\left[H_*^f(H_*^r)^{-1}[-\nabla \mathcal{L}^f(\theta_t;\varepsilon_t)]\right]. \qquad (A31)$$

$$\square$$

## A.4 Proof of Proposition 3

**Proposition 3.** *For implicit Hessian approximation in* (5)*, suppose (A1) $\beta_t, \delta_t$ is small, $\beta_t < \sqrt{\delta_t/|\nabla \mathcal{L}^r(\theta_t) - [\nabla \mathcal{L}^r(\theta_t)]^2|}$, (A2) $\mathcal{L}^r$ is $\mu$-smooth, i.e., $\|\nabla \mathcal{L}^r(\theta) - \nabla \mathcal{L}^r(\theta')\|_2 \le \mu\|\theta - \theta'\|_2$, and (A3) there exist an $\zeta_t$-neighborhood $\mathcal{N}(\theta_t^r, \zeta_t)$ of the optimal model parameter $\theta_t^r = \arg\min_{\theta_t^f} \mathcal{L}^r(\theta_t^f)$, which includes $\theta_t$ and $\theta_t^f$. Then, the iterative update term approximately is,*

$$\theta_t - \theta_t^r :\approx \beta_t^2 \left[\nabla^2 \mathcal{L}^r(\theta_t)\right]^{-1} \nabla \mathcal{L}^u(\theta_t) = \beta_t^2(H_t^r)^{-1}\nabla \mathcal{L}^u(\theta_t) \qquad (A32)$$

*Proof.* The objective function of implicit Hessian approximation can be formulated as:

$$\min_{\theta_t^f} \mathcal{L}^r(\theta_t^r) \quad \text{s.t.} \quad \theta_t^f = \theta_t - \beta_t \nabla \mathcal{L}^u(\theta_t). \qquad (A33)$$

We need to get the optimal parameter $\theta_t^r$ that minimizes (A33), which means $0 = \frac{\partial \mathcal{L}^r(\theta_t^r)}{\partial \theta_t^r}$. We can take the Taylor expansion at $\theta_t$,

$$0 = \frac{\partial \mathcal{L}^r(\theta_t^r)}{\partial \theta_t^r} = \nabla \mathcal{L}^r(\theta_t^r) = \nabla \mathcal{L}^r(\theta_t) + H_t^r(\theta_t^r - \theta_t) + (\theta_t^r - \theta_t)^\top \otimes \mathbf{T} \otimes (\theta_t^r - \theta_t) + o(\zeta_t) \qquad (A34)$$

where $H_t^r = \nabla^2 \mathcal{L}^r(\theta_t)$ and $\mathbf{T}$ represent the Hessian matrix and the third-order symmetric tensor on the remaining set, respectively, and $\otimes$ denotes the Kronecker product.

From (**A2**) and (**A3**), we can reduce the first-order term to $o(\mu\zeta_t)$,

$$\|\nabla \mathcal{L}^r(\theta_t^r) - \nabla \mathcal{L}^r(\theta_t)\|_2 \le \mu\|\theta_t^r - \theta_t\|_2 \le \mu\zeta_t. \qquad (A35)$$

To simplify the second-order term with (**A1**), we have

$$\begin{aligned}
&(\theta_t^r - \theta_t)^\top \otimes \mathbf{T} \otimes (\theta_t^r - \theta_t) \\
=&(\theta_t^f - \theta_t)^\top \otimes \mathbf{T} \otimes (\theta_t^f - \theta_t) + o(\epsilon_t) \\
=&\mathbf{C} \odot (\theta_t^f - \theta_t)^2 + o(\epsilon_t) \\
\approx&\beta^2 (\nabla \mathcal{L}^u(\theta_t))^2 + o(\epsilon_t) \\
=&\beta^2 \nabla \mathcal{L}^u(\theta_t) + o(\delta_t) + o(\epsilon_t)
\end{aligned} \qquad (A36)$$

Bringing (A35) and (A36) into (A34), we have

$$0 = \nabla \mathcal{L}^r(\theta_t^r) \approx H_t^r(\theta_t^r - \theta_t) + \beta^2 \nabla \mathcal{L}^u(\theta_t) + o(\delta_t) + o(\epsilon_t) + o(\mu\epsilon_t) + o(\epsilon_t^2) \qquad (A37)$$

Then, we can derive

$$\theta_t - \theta_t^r \approx \beta_t^2(H_t^r)^{-1}\nabla \mathcal{L}^u(\theta_t). \qquad (A38)$$

$$\square$$

# B  Related Works

## B.1  Related Works on Machine Unlearning

**Data Forgetting.** MU is driven by the imperative to remove the influence of specific data from pre-trained models [2, 3, 18, 64], intrinsically linked to differential privacy [18, 64, 65, 15, 66], which aims to enhance the privacy protection of training data. Exact MU, which is approached from a parameter probability perspective, has been thoroughly explored within convex optimization problems and linear models [18, 15, 39, 67, 16]. These studies have established methods allowing models to forget data exactly while adhering to a specified privacy budget, thus significantly reducing the risk of privacy attacks [68–72]. However, in deep learning models, exact data forgetting typically requires retraining from scratch [1], a process whose computational intensity makes it impractical for routine application. This challenge underscores an urgent need for developing more efficient unlearning techniques that do not compromise the model's utility.

**Gradient-based Approximate Machine Unlearning.** To enhance data forgetting efficiency, research has honed in on aligning the forgetting objective with the model's output probability distribution, termed approximate MU [3, 21]. Numerous studies [22, 19, 38, 20, 21, 23, 26, 73] have developed specialized loss functions to prompt the model to expunge specific data, aiming to mitigate the risks of privacy breaches such as membership inference [74, 68–72] and data reconstruction attacks [75, 76]. Echoing the phenomenon of catastrophic forgetting observed in continual learning [77–79], training on forgetting data has precipitated significant declines in overall performance on remaining data. Various strategies [26, 24, 25] have emerged to uphold the model's original generalization capabilities, predominantly through fine-tuning the remaining set. Our approach provides a comprehensive analysis of iterative strategies employed in gradient-based approximate MU methods and introduces curvature information from the remaining set to better preserve the model's generalization capabilities.

**Machine Unlearning for Generative Models.** Recent advancements in text-conditional image generation models have demonstrated remarkable capability in producing images that accurately reflect textual descriptions [40, 42, 41, 58]. Despite these achievements, extensive research [6, 12, 80] has underscored significant security and privacy concerns associated with these technologies. The mechanisms underlying these issues are not yet fully understood. In response, there is a critical demand for the development of MU methods to bolster the trustworthiness of these models, facilitating their wider adoption. While pioneering studies [12, 14, 11, 13] have begun to address concept deletion within diffusion models, the dual objectives of maintaining generalization and ensuring efficient data forgetting continue to pose significant challenges.

## B.2  Related Works on Steepest Descent in Optimization

**Steepest Descent and Natural Gradient.** Steepest Descent [81, 29, 27, 30, 82] is one of the foundational algorithms in optimization, particularly in the context of machine learning and neural networks. Despite its simplicity and widespread use, Steepest Descent can suffer from slow convergence, especially in ill-conditioned problems where the objective function's curvature varies significantly across different dimensions [29]. Natural Gradient [83, 30, 84] is proposed as an enhancement of the standard gradient descent that addresses some of its limitations by considering the underlying geometry of the parameter space. Natural Gradient employs the Fisher Information Matrix to scale the gradient adaptively, leading to more efficient optimization. Natural Gradient has been shown to significantly accelerate convergence and improve optimization performance in various applications, including deep learning [30, 27] and continual learning [28]. We get inspiration from these techniques and try to improve MU update direction with preserved set curvature information.

**Hessian Approximation.** Computing the exact Hessian matrix is often impractical for large-scale problems due to its computational and memory requirements. To address this, various Hessian Approximation methods have been developed. Notable approaches include Diagonal and Low-rank Approximations [34, 85, 86], Stochastic Approximation [87–89], and Quasi-Newton Methods [90, 91]. [50] reveals that the regularization-based method is an explicit approximation of the Hessian, while the meta-learning method is an implicit estimation of it. By leveraging second-order information in a computationally feasible manner, these techniques strike a balance between accuracy and efficiency, facilitating the optimization of large-scale problems.

### B.3 Preliminary on Diffusion Model

**Conditional Diffusion Models and Classifier-free guidance.** Diffusion models [40] have gained prominence in the field of generative modeling, particularly for their effectiveness in generating high-quality images. A sample $x_T$ is sampled from a Gaussian distribution and gradually denoised for $T$ time steps, finally recovering a clean sample $x_0$. Conditional Diffusion Model [40] is a variant that conditions the generation process on additional information $c$ such as class labels, text descriptions, or other modalities. In practice, the conditional diffusion model is trained to predict the noise $\epsilon_\theta(x_t|c)$ to form the denoising process $p_\theta(x_{t-1}|x_t, c)$, where $x_t$ is a noisy version of the input $x$. The corresponding objective of the conditional diffusion model is typically formulated as:

$$\ell_{\mathrm{MSE}}(\theta; x, c) = \mathbb{E}_{t, \epsilon \sim \mathcal{N}(0,1)} \left[ \|\epsilon - \epsilon_\theta(x_t|c)\|_2^2 \right], \tag{A39}$$

with $t$ uniformly sampled from $\{1, \ldots, T\}$. In this setting, classifier-free guidance [41] is proposed to encourage the sampling procedure to find $x$ with high $\log p(c|x)$. Then diffusion process is given by $\hat{\epsilon}_\theta(x_t|c) = (1-w)\epsilon_\theta(x_t|\emptyset) + w\epsilon_\theta(x_t|c)$, where $\hat{\epsilon}_\theta(x_t|c)$ represents the noise estimation obtained by the conditional diffusion model given $c$, $w \in [0, 1]$ is the guidance weight, $\emptyset$ denotes the 'null' condition. The generation process initiates with Gaussian noise $\hat{x}_T \sim \mathcal{N}(0, 1)$ and repeats denoising the data by $\hat{\epsilon}_\theta(\hat{x}_t|c)$ to obtain $\hat{x}_{t-1}$ until $t = 0$, producing the authentic data conditioned on $c$.

**Latent Diffusion Models.** Direct training of conditional diffusion models in high-resolution pixel space is often computationally prohibitive. Latent diffusion models (LDMs) [42, 58] address this challenge with an image compression approach. Specifically, an autoencoder is trained using perceptual loss and a patch-based adversarial objective to master the process of perceptual image compression. The input images can be embedded into smaller latent representations with the learned encoder $E$. The trained autoencoder facilitates the transition to a low-dimensional latent space where the diffusion model is more efficiently trained on the representations $z = E(x)$ rather than on high-resolution images $x$. Authentic images can then be generated by sampling a representation $z$ from the diffusion model and subsequently reconstructed into an image with the learned decoder $x = D(z)$.

## C  Algorithm

We present the algorithm of our proposed SFR-on in **Alg. A1**. In steps 3-4, we first calculate the forget-remain balanced weight saliency mask. In steps 6-8, we compute the adaptive coefficients to weight the forgetting gradient ascent, followed by one step of forgetting update within the inner loop. In steps 9-13, we fine-tune the model on the remaining set. In step 14, we perform the slow weight update for the model in the outer loop.

## D  Evaluation Metrics

### D.1  Evaluation Metrics for Image Classification

● Forgetting accuracy (**FA**): FA is the accuracy of the unlearned model on the forgetting dataset $\mathcal{D}^f$. *A favorable approximate MU method should reduce the disparity of FA with the retrained model.*

● Remaining accuracy (**RA**): RA is the accuracy of the unlearned model on the remaining dataset $\mathcal{D}^r$. *A favorable approximate MU method should reduce the disparity of RA with the retrained model.*

● Testing accuracy (**TA**): TA is the accuracy of the unlearned model on the testing dataset $\mathcal{D}^t$. $\mathcal{D}^t$ in random subset forgetting is sampled from the same distribution as the remaining dataset, while in class-wise forgetting $\mathcal{D}^t$ excludes the samples from the forgetting class. *A favorable approximate MU method should reduce the disparity of TA with the retrained model.*

● Membership inference attack success rate on $\mathcal{D}^f$ (**MIA**): Following [33], we use a prediction entropy-based membership inference attack to evaluate the privacy preservation of the unlearned model. We first need to train an adversarial classifier to predict whether or not a particular example was contained in the training dataset. The prediction of the remaining dataset and the testing dataset by the unlearned model is collected to compute the label-agnostic prediction entropy for the attack classifier training. Specifically, a Logistic Regression classifier is trained on the remaining prediction entropy labeled as '1' and the testing prediction entropy labeled as '0'. Then, this attack classifier is

---

**Algorithm A1** The Algorithm of Proposed SFR-on

---

1: **Input:** Forgetting set $\mathcal{D}^f$, remaining set $\mathcal{D}^r$, pre-trained model weights $\theta$, outer loop learning rate $\alpha$, inner loop iteration number $T_{\text{in}}$, outer loop iteration number $T_{\text{out}}$, initial inner loop learning rate for forgetting $\beta^f$, initial inner loop learning rate for remaining $\beta^r$, temperature scalar $\lambda$, weight saliency mask threshold $\gamma$

2: **Initialize:** $\theta_0 = \theta$.

3: Compute $F^r_{\text{diag}}, F^f_{\text{diag}}$ by (10).

4: Compute weight saliency mask by $\mathbf{m} = \mathbf{I}[F^f_{\text{diag}}(F^r_{\text{diag}})^{-1} \geq \gamma]$.

5: **for** $t = 1$ **to** $T_{\text{out}}$ **do**

6:     Sample forgetting sample batch $B^f_t$ **from** $\mathcal{D}^f$

7:     Compute adaptive coefficients $\tilde{\varepsilon}_{t-1}$ by (9).

8:     $\theta^f_{t-1} = \theta_{t-1} - \beta^f \nabla \mathcal{L}^f(\theta_{t-1}; \tilde{\varepsilon}_{t-1})$

9:     $\theta^r_0 = \theta^f_{t-1}$

10:     **for** $t' = 1$ **to** $T_{\text{in}}$ **do**

11:         Sample remaining sample batch $B^r_t$ **from** $\mathcal{D}^r$

12:         $\theta^r_{t'} = \theta^r_{t'-1} - \beta^r \nabla \mathcal{L}^r(\theta^r_{t'-1})$

13:     **end for**

14:     $\theta_t = \theta_{t-1} - \alpha(\theta_{t-1} - \theta^r_{T_{\text{in}}})$

15: **end for**

---

applied to the forgetting prediction entropy to predict the membership of these forgetting samples. The success rate of membership inference attack on $\mathcal{D}^f$ is quantified by the true positive rate predicted by our classifier, MIA $= TP/N^f$, where $TP$ represents the count of forgetting samples still identified as training samples and $N^f$ is the size of the forgetting dataset. Due to the limitations of the membership inference attack, the attack based on a linear classifier is weak and fails to distinguish all forgetting samples predicted by the retrained model as test samples. The more advanced shadow model-based sample-wise attack is too time-consuming to evaluate the MU method. Therefore, we only regard MIA as a readout function of the forgetting effect and advocate for the development of more precise and efficient membership inference attack techniques to enhance MU method evaluation. Note that *a favorable approximate MU method also should reduce the disparity of MIA with the retrained model.*

● Output KL divergence with the retrained model ($D_{\text{KL}}$): Given that it is impractical to traverse all sample spaces, we actually calculate the empirical output KL divergence with RT $\theta_*$. We first collect the predicted class probabilities from both the unlearned and retrained models across the remaining and forgetting datasets. Then, we empirically compute the output KL divergence as follows:

$$D_{\text{KL}} = \frac{1}{N^r + N^f} \left( \sum_{i \in \mathcal{D}^r} \sum_{c \in \mathcal{C}} p(z^r_{i,c}; \theta_*) \log \frac{p(z^r_{i,c}; \theta_*)}{p(z^r_{i,c}; \theta_u)} + \sum_{j \in \mathcal{D}^f} \sum_{c \in \mathcal{C}} p(z^f_{j,c}; \theta_*) \log \frac{p(z^f_{j,c}; \theta_*)}{p(z^f_{j,c}; \theta_u)} \right),$$

(A40)

where $\mathcal{C}$ denotes the set of the prediction classes, $\theta_u$ is the unlearned model parameter, and $z^r_{i,c}$ and $z^f_{j,c}$ represent the $i$-th remaining sample posterior of the class $c$ and the $j$-th forgetting sample posterior of the class $c$, respectively. Note that even the output KL divergence between retrained models is not zero due to the randomness of the training algorithm. *A favorable approximate MU method is hoped to display as low output KL divergence with RT as possible.*

● Run-time efficiency (**RTE**): This measures the computation efficiency of an MU method. We record RTE in *minutes. MU methods are more efficient the lower their RTE are.* Clearly, RT is not an efficient MU method.

### D.2 Evaluation Metrics for Image Generation

● Forgetting accuracy (**FA**): We additionally train models to classify the images generated by the unlearned generative model. For CIFAR-10, following [35], we use ResNet-34 from `torchvision` pre-trained on ImageNet and fine-tune it on CIFAR-10 for 20 epochs. For ImageNet, we directly use pre-trained ViT-L [92] from `torchvision` as the classifier. We compute the classification accuracy

of 500 images generated by the unlearned model on each forgetting category as FA. *A favorable approximate MU method is hoped to exhibit as low FA as possible.*

• Fréchet Inception Distance (**FID**): We evaluate the remaining image fidelity of the unlearned model by assessing the standard image generation metrics FID. The model performed class-wise forgetting of CIFAR-10 generates 1000 images for each remaining class for FID. We sample 10000 images from the unlearned DiT on ImageNet by randomly selecting the remaining classes to calculate FID. *A favorable approximate MU method is expected to achieve as high FID as possible.*

# E    Implementation Details

## E.1    Baselines

**MU methods for image classification**:

- **FT** [22, 19, 38]: It fine-tunes the pre-trained model only on the remaining dataset to obtain the unlearned model. The code source is `https://github.com/OPTML-Group/Unlearn-Sparse`.

- **GA** [20, 21]: It conducts gradient ascent for the pre-trained model only on the forgetting dataset to obtain the unlearned model. The code source is `https://github.com/OPTML-Group/Unlearn-Sparse`.

- **RL** [20]: It replaces the forgotten set label with a random label that is not the same as the original label. Then, the modified forgetting set and the retained set are combined for fine-tuning the pre-trained model with the cross-entropy loss. The code source is `https://github.com/OPTML-Group/Unlearn-Saliency`.

- **SalUn** [26]: It first obtained the top-$k$% large salient weight mask sorted by the absolute value of parameter gradient to the forgetting set loss. The pre-trained model is then fine-tuned using the same pipeline as for RL, and the gradient is modified using the weight saliency mask so that only the top-$k$% significant parameters are optimized. The code source is `https://github.com/OPTML-Group/Unlearn-Saliency`.

- **BT** [23]: It stores pre-trained and randomly initialized models as competent and incompetent teachers, respectively. The unlearned model is acquired by minimizing the output KL divergence of the pre-trained model with the incompetent teacher on the forgetting set and with the competent teacher on the remaining set. The code source is `https://github.com/vikram2000b/bad-teaching-unlearning`.

- **SCRUB** [25]: It only saves the pre-trained model as the teacher model. Then, it optimizes the pre-trained model to minimize the KL divergence with the teacher on the remaining set and maximize the KL divergence with the teacher on the forgetting set. The cross-entropy loss of the remaining set is added to further maintain the performance. The code source is `https://github.com/meghdadk/SCRUB`.

**MU methods for image generation**:

- **SA** [35]: It needs to generate replaying remaining samples using pre-trained models in advance and calculate fisher diagonal on the replaying remaining samples for modulating parameter regularization terms. Its unlearning loss is composed of three components: the MSE loss to make the output generated by the pre-trained model for forgetting classes or concepts close to random noise, the generative model MSE loss optimized on the replaying remaining samples, and a parameter regularization term to maintain the pre-trained model parameters with second-order curvature modulation of Fisher diagonal. The code source is `https://github.com/clear-nus/selective-amnesia`.

- **ESD** [11]: It saves a frozen SD as a copy. The SD model is then optimized so as to push the condition score for the erasing concept far away from the corresponding condition score generated by the frozen SD, and align with the unconditioned score generated by the copy. The code source is `https://github.com/rohitgandikota/erasing`.

Table A1: Summary of hyperparameters for each method on unlearning 10% random subset of CIFAR-10 in **Tab. 2**. 'lr' is short for learning rate.

| Methods | Hyperparameters |
|---|---|
| Pretrain | epoch = 200, cosine scheduler, lr = 0.1 |
| RT | epoch = 200, cosine scheduler, lr = 0.1 |
| FT | epoch = 10, cosine scheduler, lr $= 10^{-2}$ |
| GA | epoch = 15, constant scheduler, lr $= 3 \times 10^{-4}$ |
| RL | epoch = 10, cosine scheduler, lr $= 2 \times 10^{-3}$ |
| SalUn | epoch = 10, cosine scheduler, lr $= 3 \times 10^{-4}$, threshold = top-20% |
| BT | epoch = 10, cosine scheduler, lr $= 3 \times 10^{-3}$, temperature scalar = 1.0 |
| SCRUB | epoch = 10, constant scheduler, lr $= 3 \times 10^{-4}$, temperature scalar = 4.0, $\alpha = 0.001$ |
| SFR-on | $T_{\text{out}} = 1500$, cosine scheduler, $\alpha = 1.0, T_{\text{in}} = 5, \beta^f = 0.25, \beta^r = 0.01, \lambda = 0.5, \gamma = 1$ |

Table A2: Summary of hyperparameters for each method on unlearning 10% random subset of TinyImageNet in **Tab. 2**. 'lr' is short for learning rate.

| Methods | Hyperparameters |
|---|---|
| Pretrain | epoch = 10, cosine scheduler, lr $= 10^{-4}$ |
| RT | epoch = 10, cosine scheduler, lr $= 10^{-4}$ |
| FT | epoch = 1, cosine scheduler, lr $= 10^{-4}$ |
| GA | epoch = 9, constant scheduler, lr $= 3 \times 10^{-6}$ |
| RL | epoch = 1, cosine scheduler, lr $= 10^{-4}$ |
| SalUn | epoch = 1, cosine scheduler, lr $= 10^{-4}$, threshold = top-50% |
| BT | epoch = 1, cosine scheduler, lr $= 10^{-4}$, temperature scalar = 1.0 |
| SCRUB | epoch = 1, constant scheduler, lr $= 10^{-4}$, temperature scalar = 4.0, $\alpha = 0.001$ |
| SFR-on | $T_{\text{out}} = 500$, cosine scheduler, $\alpha = 1.0, T_{\text{in}} = 1, \beta^f = 0.01, \beta^r = 2 \times 10^{-5}, \lambda = 0.6, \gamma = 1$ |

## E.2 Training Details for Image Classification

For **CIFAR-10**, **CIFAR-100**, and **SVHN** using **ResNet-18**, all methods use the SGD optimizer with momentum of 0.9, weight decay of $5 \times 10^{-4}$, and batch size of 128. Our SFR-on train 1500 steps with the constant outer loop learning rate of $\alpha = 1.0$, inner loop iteration number $T_{\text{in}} = 5$. SFR-on search inner loop learning rate for forgetting in range $[0.1, 0.5]$ and for remaining in range $[10^{-3}, 10^{-2}]$, temperature scalar $\lambda$ in range $[0.0, 2.0]$, and threshold $\gamma$ in list $[0.3, 1.0, 3.0, 10.0]$. Experiments are run on 1 RTX 4090. A summary of the hyperparameters for each method is shown in **Tab. A1**.

For **TinyImageNet**, **Swin-T** is initialized from `torchvision` weight pre-trained on ImageNet. All methods use the AdamW optimizer [93] with weight decay of 0.05 and batch size of 128. Our SFR-on train 500 steps with the constant outer loop learning rate $\alpha = 1.0$, inner loop iteration number $T_{\text{in}} = 1$. SFR-on search inner loop learning rate for forgetting in range $[0.001, 0.1]$ and for remaining in range $[10^{-5}, 10^{-4}]$, temperature scalar $\lambda$ in range $[0.0, 2.0]$, and threshold $\gamma$ in list $[0.3, 1.0, 3.0, 10.0]$. Experiments are run on 1 RTX 4090. A summary of the hyperparameters for each method is shown in **Tab. A2**.

## E.3 Training Details for Image Generation

For **CIFAR-10**, following [35], we use **DDPM**[1] based on U-Net architecture with 1000 timesteps for linear $\beta$ schedule. All methods use Adam optimizer with a constant learning rate of $10^{-4}$ and batch size of 128. Pretrain and RT train for 800K steps. SA generates 5000 images as the remaining set for replaying and calculating the Fisher diagonal, and trains for 20K steps with $\lambda = 10$ for the regularization term. SalUn obtains top-50% salient weight mask and trains for 1K steps with $\alpha = 10^{-3}$ to balance the optimization of forgetting and remaining. Our SFR-on trains for 50 steps

---
[1]https://github.com/ermongroup/ddim

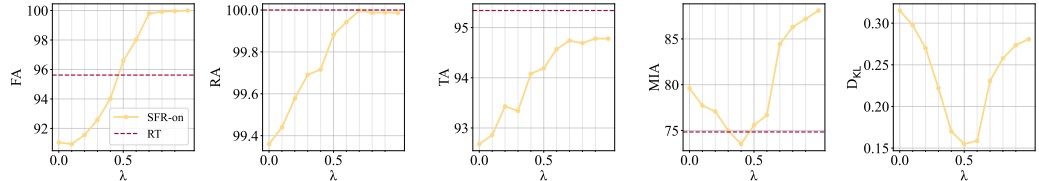

Figure A1: Performance of SFR-on with different $\lambda$ in adaptive coefficients $vs$ RT on CIFAR-10 using ResNet-18. The settings and metrics follow **Tab.** 2. The points closer to RT and with lower $D_{\mathrm{KL}}$ are better.

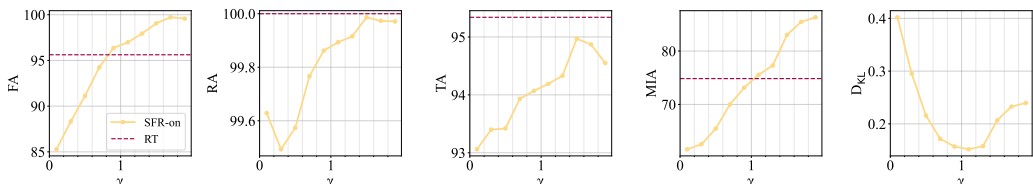

Figure A2: Performance of SFR-on with different $\gamma$ in weight saliency mask $vs$ RT on CIFAR-10 using ResNet-18. The settings and metrics follow **Tab.** 2. The points closer to RT and with lower $D_{\mathrm{KL}}$ are better.

with $T_{\mathrm{in}} = 1, \alpha = 1.0, \beta^f = 10^{-3}, \beta^r = 10^{-4}, \lambda = 0.5, \gamma = 3$. Experiments are run on 2 RTX 4090s.

For **ImageNet**, we use pre-trained **DiT-XL/2**[2] with $256 \times 256$ resolution. All methods use AdamW optimizer with a constant learning rate of $10^{-4}$ and batch size of 1. SA calculates the Fisher diagonal on randomly sampled 2000 remaining data, and trains for 10K steps with $\lambda = 10$ for the regularization term. SalUn obtains top-50% salient weight mask and trains for 10K steps with $\alpha = 10^{-3}$ to balance the optimization of forgetting and remaining. Our SFR-on trains for 500 steps with $T_{\mathrm{in}} = 1, \alpha = 1.0, \beta^f = 10^{-7}, \beta^r = 10^{-4}, \gamma = 3$. Since the batch size is 1, we ignore $\lambda$ in adaptive coefficients. Experiments are run on 1 RTX 4090.

### E.4 Training Details for Concept Forgetting

For **concept forgetting of 'nudity'**, following [26], we use **SD V1.4**[3] to generate 1K images with the prompt 'a photo of a nude person' as the forgetting set and additional 1K images with the prompt 'a photo of a person wearing clothes' as the remaining set. All methods use Adam optimizer with a constant learning rate of $10^{-5}$ and batch size of 1. ESD trains for 1K steps with negative guidance of 1.0. SalUn trains for 1K steps with 50% sparsity weight saliency and $\alpha = 0.1$. Our SFR-on trains for 200 steps with $T_{\mathrm{in}} = 1, \alpha = 1.0, \beta^f = 10^{-6}, \beta^r = 10^{-5}, \gamma = 3$. Since the batch size is 1, we ignore $\lambda$ in adaptive coefficients. Experiments are run on 1 RTX 4090.

## F  Additional Experimental Results

### F.1  Ablation Study on Temperature Scalar $\lambda$ and Threshold $\gamma$

In our SFR-on method, we investigate the impact of two key hyperparameters, temperature scalar $\lambda$ for adaptive coefficients and threshold $\gamma$ for the weight saliency mask, on the unlearning performance in image classification. We vary $\lambda$ within the range $[0, 1]$ and $\gamma$ within $[0, 2]$. The results in **Fig.** A1 and A2 indicate that increasing $\gamma$ and $\lambda$ enhances model retention capabilities, but it could adversely affect both the forgetting performance and privacy protection. Therefore, we select the hyperparameters for our method based on the goal of minimizing the output KL divergence from Retrain, aligning with the objectives of approximate MU.

---

[2]https://github.com/facebookresearch/DiT
[3]https://huggingface.co/CompVis/stable-diffusion-v1-4

Table A3: MU performance for unlearning 50% random subset of CIFAR-10 using ResNet-18. The content format follows **Tab. 2**.

| Methods | CIFAR-10 Random Subset Forgetting (50%) | | | | Avg.D ↓ | $D_{\text{KL}}$ ↓ | RTE |
|---|---|---|---|---|---|---|---|
| | FA | RA | TA | MIA | | | |
| RT | $93.36_{\pm 0.10}$ (0.00) | $100.00_{\pm 0.00}$ (0.00) | $93.11_{\pm 0.29}$ (0.00) | $69.02_{\pm 0.01}$ (0.00) | 0.00 | 0.10 | 138.95 |
| FT | $99.95_{\pm 0.03}$ (6.59) | $100.00_{\pm 0.00}$ (0.00) | $94.57_{\pm 0.19}$ (1.46) | $88.97_{\pm 0.39}$ (19.95) | 7.00 | 0.44 | 6.63 |
| GA | $97.25_{\pm 0.23}$ (3.89) | $97.25_{\pm 0.23}$ (2.75) | $90.48_{\pm 0.30}$ (2.63) | $82.89_{\pm 0.00}$ (13.77) | 5.76 | 0.42 | 2.43 |
| RL | $99.04_{\pm 0.19}$ (5.67) | $99.05_{\pm 0.19}$ (0.95) | $93.79_{\pm 0.20}$ (0.68) | $71.87_{\pm 0.01}$ (2.85) | 2.54 | 0.55 | 5.15 |
| SalUn | $99.42_{\pm 0.14}$ (6.06) | $99.38_{\pm 0.16}$ (0.62) | $94.25_{\pm 0.15}$ (1.14) | $70.35_{\pm 0.01}$ (1.32) | 2.29 | 0.55 | 7.24 |
| BT | $99.40_{\pm 0.11}$ (6.04) | $99.38_{\pm 0.11}$ (0.62) | $94.24_{\pm 0.14}$ (1.13) | $72.48_{\pm 0.00}$ (3.46) | 2.81 | 0.54 | 6.89 |
| SCRUB | $84.65_{\pm 14.01}$ (8.73) | $85.56_{\pm 13.79}$ (14.44) | $81.71_{\pm 11.83}$ (11.41) | $64.03_{\pm 0.14}$ (5.31) | 9.97 | 0.50 | 2.73 |
| SFR-on | $93.50_{\pm 0.33}$ (0.13) | $97.42_{\pm 0.32}$ (2.58) | $91.12_{\pm 0.25}$ (1.99) | $68.31_{\pm 0.01}$ (0.71) | 1.35 | 0.18 | 3.57 |

Table A4: MU performance for unlearning 10% random subset of CIFAR-100 using ResNet-18. The content format follows **Tab. 2**.

| Methods | CIFAR-100 Random Subset Forgetting (10%) | | | | Avg.D ↓ | $D_{\text{KL}}$ ↓ | RTE |
|---|---|---|---|---|---|---|---|
| | FA | RA | TA | MIA | | | |
| RT | $77.96_{\pm 0.52}$ (0.00) | $99.98_{\pm 0.00}$ (0.00) | $77.23_{\pm 0.30}$ (0.00) | $41.79_{\pm 0.01}$ (0.00) | 0.00 | 0.35 | 84.21 |
| FT | $83.99_{\pm 0.30}$ (6.04) | $98.84_{\pm 0.03}$ (1.15) | $75.28_{\pm 0.26}$ (1.95) | $60.92_{\pm 0.01}$ (19.14) | 7.07 | 0.66 | 4.87 |
| GA | $70.57_{\pm 6.01}$ (7.38) | $71.53_{\pm 5.88}$ (28.45) | $51.48_{\pm 3.83}$ (25.75) | $42.39_{\pm 0.02}$ (0.60) | 15.55 | 1.43 | 0.73 |
| RL | $99.39_{\pm 0.08}$ (21.44) | $99.96_{\pm 0.01}$ (0.02) | $72.62_{\pm 0.11}$ (4.61) | $44.68_{\pm 0.00}$ (2.89) | 7.24 | 1.01 | 4.81 |
| SalUn | $99.42_{\pm 0.06}$ (21.47) | $99.96_{\pm 0.01}$ (0.02) | $73.05_{\pm 0.20}$ (4.19) | $45.28_{\pm 0.01}$ (3.49) | 7.29 | 0.99 | 4.90 |
| BT | $99.63_{\pm 0.07}$ (21.68) | $99.97_{\pm 0.00}$ (0.01) | $73.28_{\pm 0.10}$ (3.95) | $45.25_{\pm 0.00}$ (3.46) | 7.28 | 1.00 | 6.39 |
| SCRUB | $90.32_{\pm 0.23}$ (12.36) | $99.11_{\pm 0.14}$ (0.87) | $78.19_{\pm 0.10}$ (0.95) | $43.38_{\pm 0.00}$ (1.60) | 3.95 | 0.61 | 4.04 |
| SFR-on | $77.82_{\pm 0.75}$ (0.14) | $99.77_{\pm 0.02}$ (0.22) | $74.76_{\pm 0.23}$ (2.47) | $47.57_{\pm 0.00}$ (5.78) | 2.15 | 0.60 | 5.41 |

## F.2 Additional Results for Image Classification

We conducted experiments to assess the performance of unlearning a 50% random subset of CIFAR-10 and a 10% random subset on two additional datasets, CIFAR-100 and SVHN. The results in **Tab. A3, A4**, and **A5** demonstrate that our SFR-on method consistently achieves the closest average metric disparity and the smallest KL divergence relative to Retrain across all three scenarios. These findings emphasize the broad unlearning efficacy of our approach.

## F.3 Results for Natural Language Processing

The input modality of the model does not constrain our analysis or methods. Therefore, our method can seamlessly extend to other modalities beyond images, such as natural language processing using large language models (LLMs), to achieve efficient forgetting. We conduct experiments using the recently proposed benchmark of TOFU [94] fine-tuned Phi-1.5 [95] to evaluate the effectiveness of our method in the LLM unlearning task, compared with four LLM unlearning baselines: gradient descent(GA), gradient difference(GradDiff [96]), negative preference optimization(NPO [97]), and its enhanced version. The TOFU dataset comprises fictional author biographies, along with questions and answers related to the authors' attributes, which helps assess methods of data forgetting on fine-tuned LLMs.

As shown in **Tab. A6**, our method achieves superior forgetting quality, making the unlearned model almost indistinguishable from the retrained model based on the Truth Ratio distribution of the forget set. Additionally, our method efficiently preserves model utility.

## F.4 Ablation Study on SFR-on without Repairing

We conduct ablation experiments to assess the performance of our 'Sample-wise Adaptive Coefficient for Gradient Ascent (F)' and 'Forget-Remain Balanced Weight Saliency (S)' in the absence of repairing with the remaining set. The results in **Tab. A7** affirm that these components, when used independently, can still enhance baseline performance.

## F.5 Results for Removing Influence of A Single Data Point

Our method can be directly applied to the task of forgetting a single data point without additional adaptation, as we impose no assumptions to constrain the size of the forgetting set. We conduct experiments under two setups to validate the effectiveness of our method in unlearning either a single

Table A5: MU performance for unlearning 10% random subset of SVHN using ResNet-18. The content format follows **Tab. 2**.

| Methods | SVHN Random Subset Forgetting (10%) | | | | $\text{Avg.D} \downarrow$ | $D_{\text{KL}} \downarrow$ | RTE |
| | FA | RA | TA | MIA | | | |
|---|---|---|---|---|---|---|---|
| RT | $96.05_{\pm 0.14}$ (0.00) | $99.82_{\pm 0.01}$ (0.00) | $96.53_{\pm 0.10}$ (0.00) | $79.47_{\pm 0.01}$ (0.00) | 0.00 | 0.10 | 138.95 |
| FT | $98.74_{\pm 0.06}$ (2.69) | $99.61_{\pm 0.02}$ (0.21) | $96.40_{\pm 0.11}$ (0.14) | $81.88_{\pm 0.01}$ (2.41) | 1.36 | 0.13 | 6.37 |
| GA | $99.44_{\pm 0.03}$ (3.39) | $99.50_{\pm 0.03}$ (0.32) | $96.37_{\pm 0.02}$ (0.16) | $78.63_{\pm 0.00}$ (0.84) | 1.18 | 0.19 | 0.71 |
| RL | $99.45_{\pm 0.02}$ (3.40) | $99.40_{\pm 0.02}$ (0.41) | $96.13_{\pm 0.03}$ (0.40) | $77.31_{\pm 0.00}$ (2.16) | 1.59 | 1.01 | 4.81 |
| SalUn | $99.42_{\pm 0.01}$ (3.38) | $99.39_{\pm 0.01}$ (0.42) | $96.12_{\pm 0.03}$ (0.41) | $78.32_{\pm 0.00}$ (1.15) | 1.34 | 0.24 | 10.89 |
| BT | $99.47_{\pm 0.02}$ (3.42) | $99.43_{\pm 0.02}$ (0.39) | $96.18_{\pm 0.06}$ (0.36) | $78.47_{\pm 0.00}$ (1.00) | 1.29 | 0.24 | 9.09 |
| SCRUB | $99.15_{\pm 0.08}$ (3.10) | $99.45_{\pm 0.02}$ (0.37) | $96.11_{\pm 0.11}$ (0.43) | $79.24_{\pm 0.01}$ (0.23) | 1.03 | 0.15 | 5.91 |
| SFR-on | $96.84_{\pm 1.07}$ (0.79) | $97.64_{\pm 1.02}$ (2.18) | $95.77_{\pm 0.39}$ (0.76) | $79.43_{\pm 0.00}$ (0.04) | 0.94 | 0.11 | 3.55 |

Table A6: Unlearning performance of our method and three LLM unlearning baselines on forgetting 5% of authors or one author under the TOFU fine-tuned Phi-1.5 benchmark. 'Forget Quality' measures the KS-Test value of the Truth Ratio between the unlearned model and the retrained model after the target information is removed. 'Model Utility' evaluates the general performance retained by the unlearned model. RTE is recorded in minutes.

| Methods | Random Subset Forgetting (5%) | | | Random One Author Forgetting | | |
| | Forget Quality ↑ | Model Utility ↑ | RTE | Forget Quality ↑ | Model Utility ↑ | RTE |
|---|---|---|---|---|---|---|
| RT | - | 0.4984 | 34.84 | - | 0.5000 | 35.60 |
| GA | 0.0297 | 0.2946 | 4.79 | 0.9499 | 0.4783 | 0.95 |
| GradDiff | 0.3281 | 0.3840 | 5.57 | **0.9999** | 0.4957 | 1.15 |
| NPO | 0.1779 | 0.3653 | 4.93 | 0.9724 | 0.4814 | 1.42 |
| NPO+GradDiff | 0.5452 | 0.4085 | 5.72 | **0.9999** | 0.4923 | 1.54 |
| SFR-on | **0.6284** | **0.4313** | 3.33 | **0.9999** | **0.4999** | 1.01 |

data point or a task with suitable granularity: (1) randomly forgetting one sample in the classification task on CIFAR-10 and (2) forgetting the relevant information of one author in TOFU benchmark.

The results in **Tab.** A8 and A6 indicate that, under these two settings, our method and all baselines achieve effective forgetting while fully retaining general performance.

### F.6 Verification for Approximation in Practical implementations

Given that the approximation in the theoretical analysis may not hold in practical implementation, verifying the approximation of the formula in the actual operation is necessary. In the proofs of **Prop.** 1 and 2, considering the approximations of $\nabla \mathcal{L}^r(\theta_0)$, $\nabla \mathcal{L}^r(\theta_*)$, and $\nabla R(\theta_t)$, we demonstrate in **Fig.** A3 their gradient norms under practical unlearning settings, which are consistent with our theoretical assumptions. In our method, $\nabla \mathcal{L}^r(\theta_0) \approx \nabla \mathcal{L}^r(\theta_*) \approx \nabla R(\theta_t) \approx 0$, allowing us to incorporate second-order information on the retain set.

### F.7 Additional Results for Image Generation

The additional results of class-wise forgetting across all CIFAR-10 classes are presented in **Fig.** A4, A5, and A6. Our method successfully removes the semantics of the target classes without reducing the forgetting images to low-quality noise. Additionally, SFR-on maintains the generation fidelity of the remaining categories.

**Fig.** A7 and A8 display the additional outcomes of class-wise forgetting on ImageNet using DiT by our SFR-on method. The diagonal images within these figures depict the targeted forgetting classes, demonstrating the unlearning efficacy of our method. Conversely, the images off the diagonal represent various non-forgetting categories, showcasing the unlearned DiT's generalization capability preserved across extensive class conditions.

## G Broader Impacts and Limitations

**Broader Impacts.** SFR-on can enhance trustworthy deep learning and is broadly applicable across various scenarios, aligning machine learning applications with ethical human standards. SFR-on can potentially increase the fairness of machine learning systems, mitigate biases against minority groups, and promote equitable decision-making processes. Moreover, SFR-on bolsters privacy protections

Table A7: Performance of GA, our SFR-on, and ablations without using the remaining dataset, assessing unlearning 10% random subset of CIFAR-10 using ResNet-18.

| Methods | | CIFAR-10 Random Subset Forgetting (10%) | | | | | | |
|---|---|---|---|---|---|---|---|---|
| | | FA | RA | TA | MIA | Avg.D↓ | $D_{KL}$↓ | RTE |
| GA | | $93.91_{\pm1.67}$ (1.71) | $93.76_{\pm1.89}$ (6.24) | $87.00_{\pm1.64}$ (8.34) | $77.19_{\pm0.01}$ (2.35) | 4.66 | 0.36 | 0.79 |
| S F R on | | | | | | | | |
| ✓ | | $96.52_{\pm0.61}$ (0.90) | $95.48_{\pm0.69}$ (4.52) | $89.62_{\pm0.74}$ (5.72) | $76.21_{\pm0.50}$ (1.37) | 3.13 | 0.29 | 0.80 |
| ✓ ✓ | | $97.03_{\pm0.35}$ (1.41) | $96.95_{\pm0.30}$ (3.05) | $89.99_{\pm0.31}$ (5.35) | $82.18_{\pm0.36}$ (7.34) | 4.29 | 0.31 | 1.03 |
| ✓ ✓ | | $96.67_{\pm0.50}$ (1.05) | $96.78_{\pm0.64}$ (3.22) | $90.80_{\pm0.69}$ (4.54) | $74.16_{\pm0.13}$ (0.68) | 2.37 | 0.26 | 1.03 |
| ✓ ✓ ✓ ✓ | | $96.58_{\pm0.77}$ (0.96) | $99.88_{\pm0.16}$ (0.12) | $94.19_{\pm0.33}$ (1.15) | $72.26_{\pm0.01}$ (2.58) | **1.20** | **0.15** | 2.80 |

Table A8: Performance of our method and four baselines in unlearning one sample on CIFAR-10 using ResNet-18.

| Methods | CIFAR-10 Random One Sample Forgetting | | | | | | |
|---|---|---|---|---|---|---|---|
| | FA | RA | TA | MIA | Avg.D↓ | $D_{KL}$↓ | RTE |
| RT | $100.00_{\pm0.00}$ (0.00) | $100.00_{\pm0.00}$ (0.00) | $95.53_{\pm0.03}$ (0.00) | $0.00_{\pm0.00}$ (0.00) | 0.0000 | 0.0001 | 76.92 |
| FT | $100.00_{\pm0.00}$ (0.00) | $100.00_{\pm0.00}$ (0.00) | $95.47_{\pm0.11}$ (0.06) | $0.00_{\pm0.00}$ (0.00) | **0.0142** | 0.0006 | 0.38 |
| GA | $100.00_{\pm0.00}$ (0.00) | $100.00_{\pm0.00}$ (0.00) | $95.31_{\pm0.02}$ (0.22) | $0.00_{\pm0.00}$ (0.00) | 0.0553 | 0.0003 | 0.05 |
| SalUn | $100.00_{\pm0.00}$ (0.00) | $100.00_{\pm0.00}$ (0.00) | $95.47_{\pm0.00}$ (0.06) | $0.00_{\pm0.00}$ (0.00) | 0.0154 | **0.0002** | 0.12 |
| SCRUB | $100.00_{\pm0.00}$ (0.00) | $100.00_{\pm0.00}$ (0.00) | $95.64_{\pm0.03}$ (0.11) | $0.00_{\pm0.00}$ (0.00) | 0.0276 | **0.0002** | 0.11 |
| SFR-on | $100.00_{\pm0.00}$ (0.00) | $100.00_{\pm0.00}$ (0.00) | $95.60_{\pm0.05}$ (0.07) | $0.00_{\pm0.00}$ (0.00) | 0.0175 | **0.0002** | 0.10 |

within deep models and fortifies the security of these systems against potential privacy attacks. When deployed on public networks, our approach facilitates the continuous update of knowledge, enabling models to catch new developments without losing performance. In the case of generative models, implementing SFR-on reduces the likelihood of generating improper content and limits the possibility of infringing on intellectual property rights. This helps to enhance public confidence in machine learning technologies and encourages their broader acceptance and use.

**Limitations.** We acknowledge the limitations of our study and encourage further exploration. While the theoretical framework of SFR-on accommodates various input modalities, this paper does not extend evaluations to include language models or graph neural networks. Consequently, we cannot ascertain the direct applicability of our method to these modalities without adaptation. Furthermore, the absence of an asymptotic analysis for the steepest descent in machine unlearning hinders our ability to determine the disparity between our approximation method and the optimal direction, which is crucial for refining the approach. We advocate for research to address these gaps in the future.

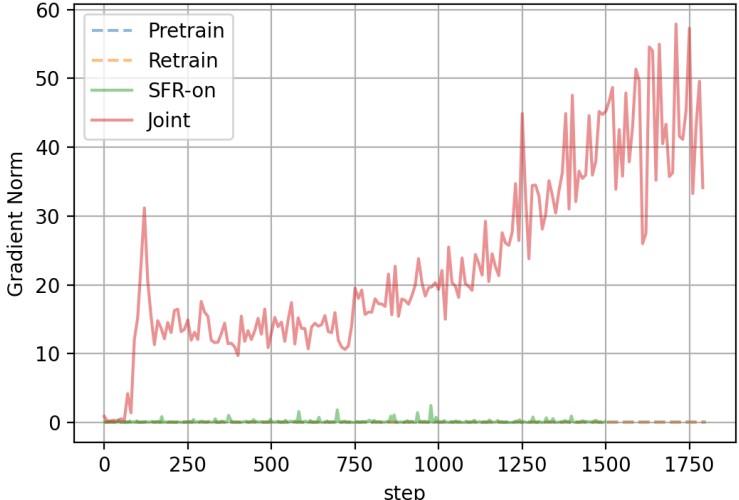

Figure A3: Gradients norm of Pretrain ($\nabla\mathcal{L}^r(\theta_0)$), Retrain ($\nabla\mathcal{L}^r(\theta_*)$), Joint ($\nabla R(\theta_t)$ in the proof of **Prop.** 1), and SFR-on ($\nabla R(\theta_t)$ in the proof of **Prop.** 2), evaluating forgetting 10% random subset of CIFAR-10 using ResNet18.

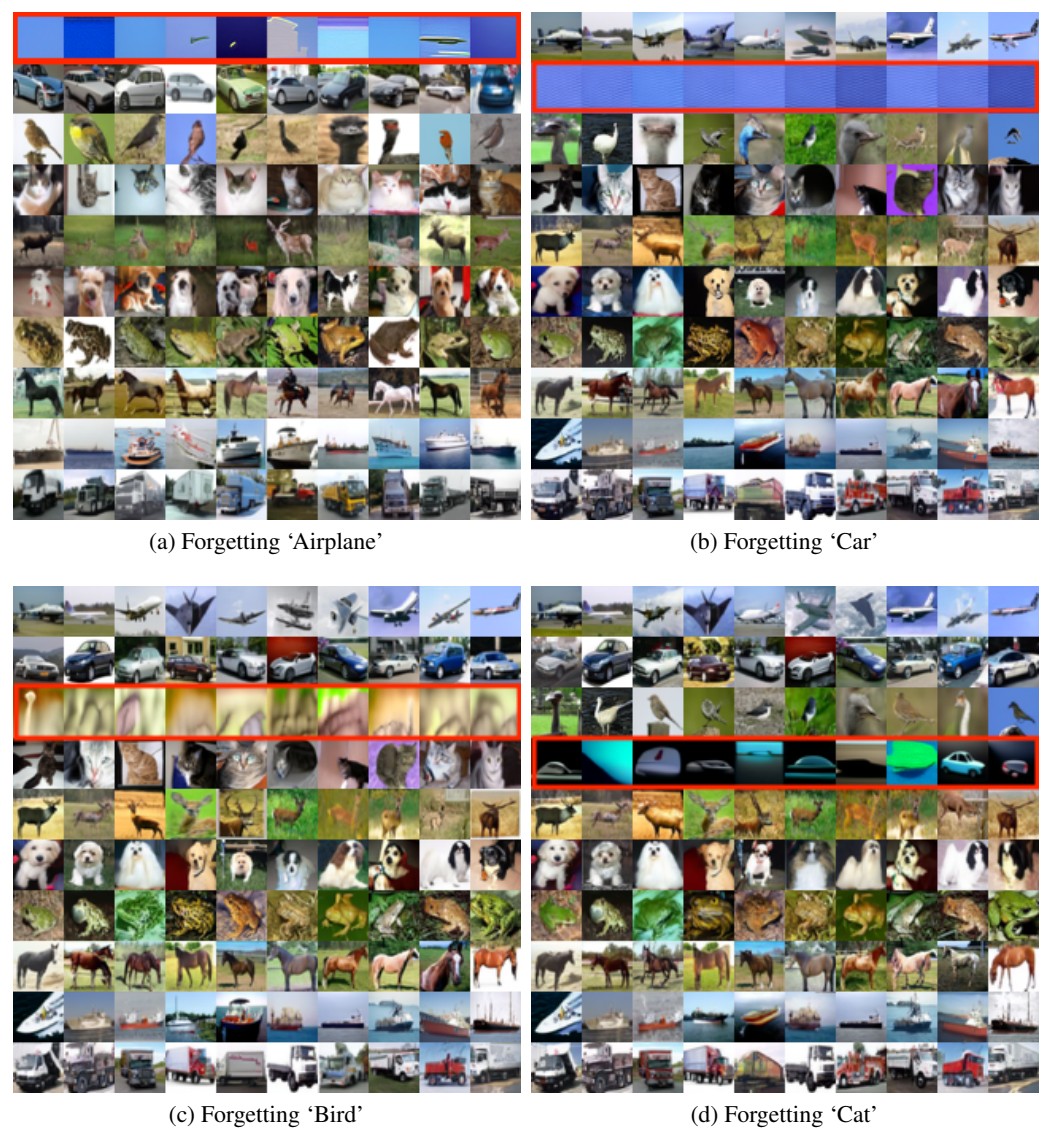

(a) Forgetting 'Airplane'

(b) Forgetting 'Car'

(c) Forgetting 'Bird'

(d) Forgetting 'Cat'

Figure A4: More class-wise unlearning results on classifier-free guidance DDPM on CIFAR-10. The forgetting class is marked with a red color. (More results will be shown in **Fig. A5** and **Fig. A6**)

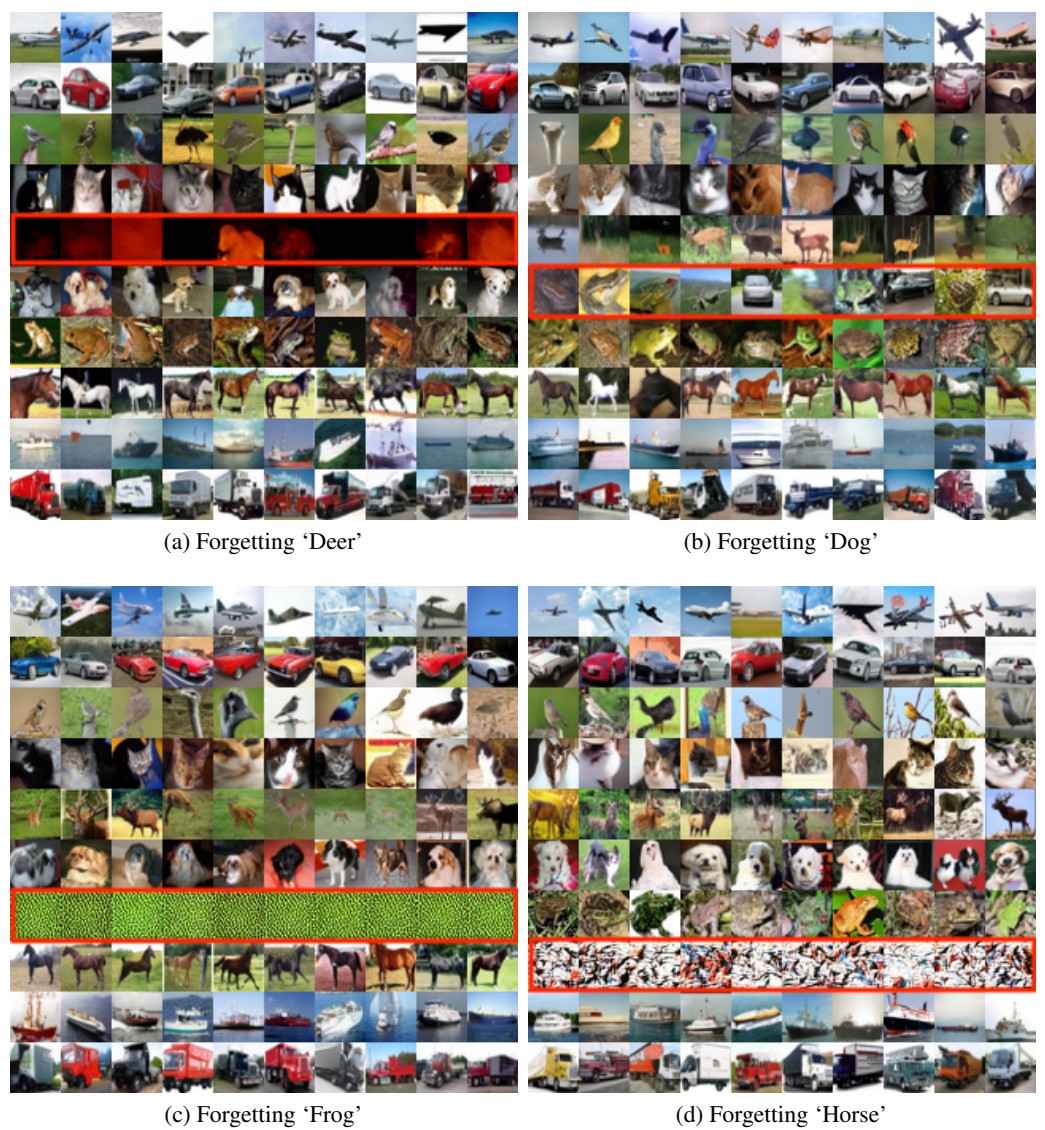

(a) Forgetting 'Deer'

(b) Forgetting 'Dog'

(c) Forgetting 'Frog'

(d) Forgetting 'Horse'

Figure A5: More class-wise unlearning results on classifier-free guidance DDPM on CIFAR-10. The forgetting class is marked with a red color (Extended results from **Fig. A4**).

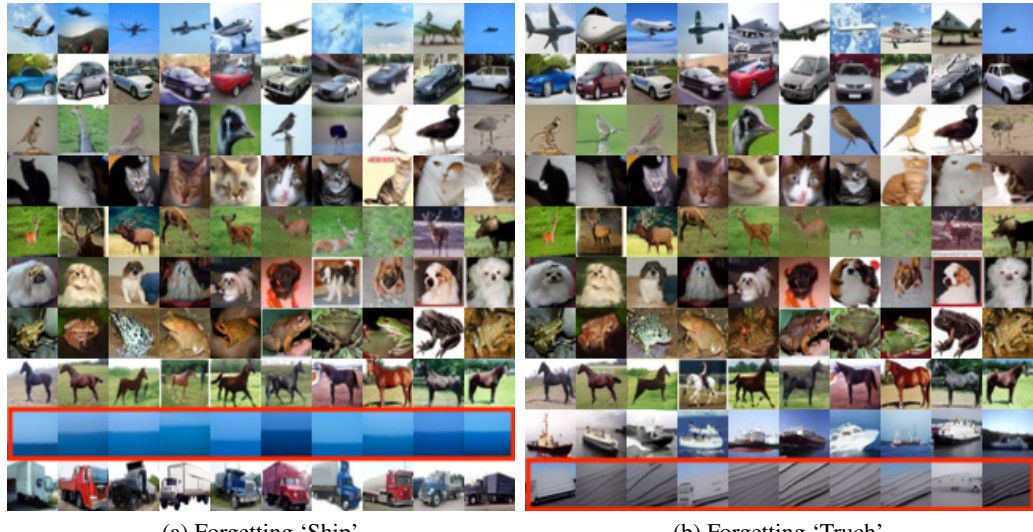

(a) Forgetting 'Ship'       (b) Forgetting 'Truch'

Figure A6: More class-wise unlearning results on classifier-free guidance DDPM on CIFAR-10. The forgetting class is marked with a red color (Extended results from **Fig. A4**).

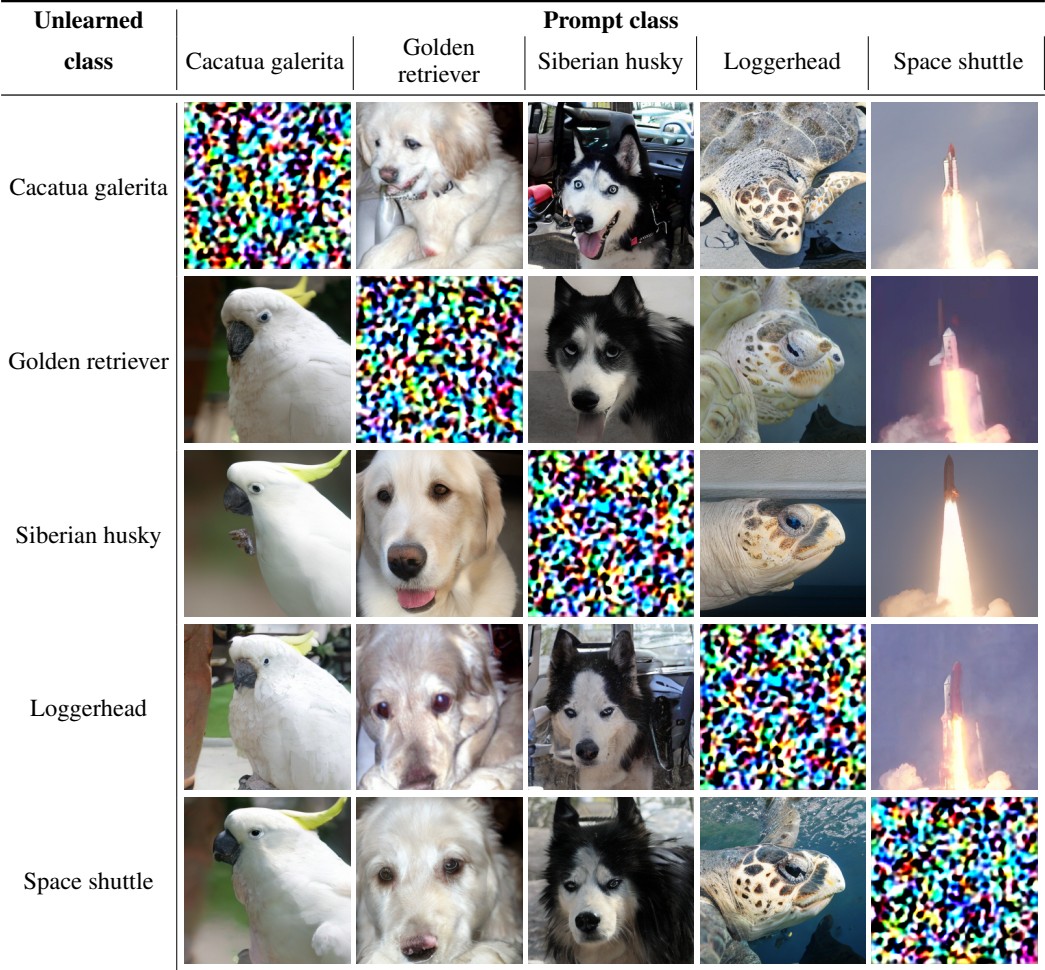

| Unlearned | Prompt class | | | | |
| class | Cacatua galerita | Golden retriever | Siberian husky | Loggerhead | Space shuttle |
|---|---|---|---|---|---|
| Cacatua galerita | | | | | |
| Golden retriever | | | | | |
| Siberian husky | | | | | |
| Loggerhead | | | | | |
| Space shuttle | | | | | |

Figure A7: Additional results of generated images using SFR-on. From the rows below, diagonal images represent the forgetting class, while non-diagonal images represent the remaining class. (More results will be shown in **Fig. A8**)

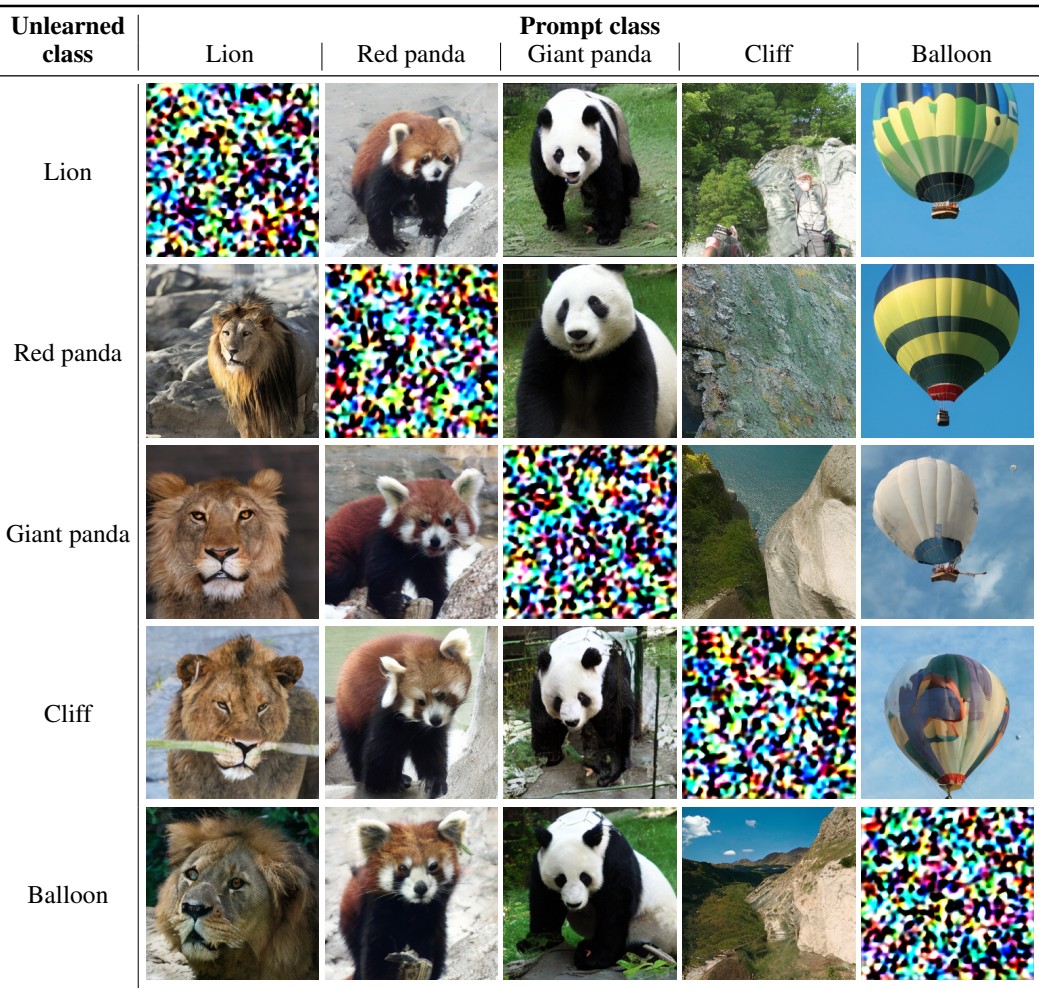

Figure A8: Additional results of generated images using SFR-on. From the rows below, diagonal images represent the forgetting class, while non-diagonal images represent the remaining class. (Extended results from **Fig. A7**)

