# OpenReview forum: "Unified Gradient-Based Machine Unlearning with Remain Geometry Enhancement"
_NeurIPS.cc/2024/Conference — NeurIPS 2024 spotlight_

### Official Review · Reviewer_S1BA · 2024-06-25

**Soundness:** 4
**Presentation:** 3
**Contribution:** 3
**Rating:** 7
**Confidence:** 2

**Summary:**

The paper tackles the problem of machine unlearning, i.e. forgetting the influence of some data points, with remain-preserving manifold geometry. Authors dig deep into the gradient based approximate unlearning methods from the perspective of steepest descent and come up with a method beyond Euclidean constraints. To overcome computationally-intensive inverse Hessian calculation, authors proposed an efficient fast-slow weight update method to approximate the Hessian-adjusted direction. The theoretical findings are experimentally validated on CIFAR and TinyImagenet image classification problem and CIFAR and ImageNet image generation problems.

**Strengths:**

1) The problem is very important to make current ML models more trustworthy and private.
2) Rigorous mathematical framework that unifies all the previous unlearning methods and introduce all the details of the proposed approach.
3) Interesting algorithm of overcoming inverse Hessian calculation by efficient fast-slow weight update.
4) Encouraging results and substantial improvements over previous baselines.
5) Easy-to-follow read with vivid illustrative figures.

**Weaknesses:**

1) Only CV unlearning is presented. If your method works on ML algorithms in general, it would be nice to present some results on other tasks such as NLP, Audio, etc.
2) One interesting observation is that machine unlearning should make the model such that "forget" and "test" set results as close as possible, meaning FA and TA should be really close like in gold-standard RT model. However the proposed method performs not really the best in terms of this metric (|FA-TA|) compared to previous approaches on TinyImageNet. I genuinely don't know if it is a good metric, I might be wrong.

Minor:

3. it would be nice to show with arrows ($\uparrow$ and $\downarrow$) in table 2 which metrics are the greater the better and which are the lesser the better.

**Questions:**

1) How does your model remove the influence of one data point, when there's a request to delete one sample from the training dataset? This might be a practical scenario.
2) Are accuracy disparities actually good measure for unlearning comparison? It might be possible that two different models predicted right different samples, thus their accuracies are the same, while they are not close to each other.
3) Why does "R-only" models have greater RTE (longer time) compared to the "full SFR" model?

**Limitations:**

Authors clearly defined the paper's broader impact and limitations in the appendix G.

---

> ### Author Rebuttal · Authors · 2024-08-06
>
> Thank you for your insightful comments, valuable feedback on our work, and constructive suggestions to help improve our presentation. We will address your concerns sequentially. **Relevant tables and figures are included in the attached rebuttal PDF**.
>
> **W1**: Including results on other modalities.
>
> > Thank you for suggesting the addition of experiments on other modalities to demonstrate the broad applicability of our method. To verify the effectiveness of our method in the context of large language models (LLM) unlearning, we conduct experiments using the recently proposed TOFU[1] benchmark, compared with four baselines[2,3]. The TOFU dataset consists of fictional author biographies, along with questions and answers related to the authors' attributes. This dataset is instrumental in evaluating methods for forgetting data information in LLMs fine-tuned on it.
> >
> >  As shown in Tab.R1, our method achieves superior forgetting quality, making the unlearned model most indistinguishable from the retrained model based on the Truth Ratio distribution of the forget set while efficiently maintaining the model utility.
> >
> > We will include this experiment of the NLP unlearning task in our Appendix, and we look forward to applying our method to more modalities, such as audio, in the future.
>
>
> **W2**: Discussion on the metric (|FA-TA|).
>
> > We greatly appreciate your insightful observation that the results of the unlearned model on the 'forget' and 'test' sets should be very close. The metric |FA-TA| can indeed serve as an indicator of the effectiveness of forgetting in the random subset unlearning task of classification. Especially considering that |FA-TA| does not need to know the retrained model. However, **|FA-TA| has certain limitations**:
> >
> > - It neglects the difference between the unlearned model and the retrained model.
> > - It has blind spots, as it might consider a randomly initialized model a good unlearned model because its |FA-TA|$\approx0$, despite the fact that such a model completely loses its utility.
> > - Narrowing it may not lead to Retrain in other unlearning tasks, such as class-wise or subclass forgetting, because |FA-TA| depends on the generalization capability to the unseen forget set.
> >
> > Overall, we think |FA-TA| is a meaingful metric for MU but is not very distinguishable. To comprehensivly evaluate MU methods, we report the Avg.D and KL divergence in Tab.2, both of which compare with the retrained model. Our method is competitive or superior to previous methods in terms of |FA-TA| and **constantly outperforms previous methods upon the overall difference and the output distribution KL divergence from the retrained model**.
>
> Minor **W3**: Comments on table presentation.
>
> > We sincerely appreciate Reviewer S1BA's meticulous review and constructive suggestions. We will add arrows to Tab.2 to enhance readability.
>
> **Q1**: Considering the scenario of removing the influence of a single data point.
>
> > We do agree that the scenario of unlearning one data point is indeed practical. **Our method can be directly applied to the task of forgetting a single data point without additional adaptation, as we do not make any assumptions to constrain the size of the forgetting set.**
> >
> > We did not consider 'removing one data point' in experiments because it is not a suitable setting for unlearning auditing:
> > - Previous literature[1,4] has shown that forgetting a single data point in deep models is a simple scenario to optimize, making it difficult to examine the performance differences of various unlearning methods.
> > - As the structure of datasets diversifies, a single sample may not be an appropriate granularity for unlearning.
> >
> > Despite the above factors, following your suggestion, we conduct experiments under two setups to verify the effectiveness of our method in unlearning either a single data point or a task with suitable granularity:
> > - Randomly forgetting one sample in the classification task on CIFAR-10 (Tab.R3).
> > - Forgetting the relevant information of one author in TOFU benchmark (Tab.R1).
> >
> > The results in Tab.R1 and Tab.R3 indicate that, under these two settings, our method and all baselines achieve effective forgetting while completely retaining general performance, making it challenging to demonstrate our improvements.
>
> **Q2**: Discussion on the metric of accuracy disparities.
>
> > We do agree with your concern on 'accuracy disparity'. In fact, evaluating the effectiveness of an unlearning method remains an open question. Accuracy disparity used in previous literature [4,5], |FA-TA| as you suggested, and output KL divergence all can be used and also have their specific limitations. Although it is not easy to find a single metric, **putting them all together, the advantage of our method is significant**.
>
> **Q3**: Why does 'R-only' cost longer time compared to 'SFR-on'?
>
> > The comparison on RTE in Tab.2 between 'R-only' and 'R-on' is fairer than with 'SFR-on' because the only difference between 'R-only' and 'R-on' is how the loss is optimized. 'R-only' combines and optimizes both forget and retain losses simultaneously, while 'R-on' uses our fast-slow weight method to alternately optimize the forget and retain losses.
> >
> > 'R-on' is faster than 'R-only' for two reasons (batch size set to $bz$):
> >
> > * In each step, 'R-on' feeds the model with $bz$ samples from either the forget or retain set, while 'R-only' combines both the forget and retain sets for a total of $2bz$ samples. Although both perform one backward operation per step, **the forward operation in 'R-only' costs longer time than in 'R-on'**.
> > * Empirically, **'R-on' requires fewer steps than 'R-only' to achieve effective forgetting**.
> >
> > 'SFR-on' adds two additional components to 'R-on'. These components are also designed with the consideration of computational efficiency, thus only slightly increasing the running time per step, and overall still less than 'R-only' (Tab.R5).

---

> ### Author Response · Authors · 2024-08-06
> **Rebuttal Reference**
>
> > [1] Maini, Pratyush et al. “TOFU: A Task of Fictitious Unlearning for LLMs.” ArXiv. 2024.
> >
> > [2] Liu, B. et al. “Continual Learning and Private Unlearning.” ArXiv. 2022.
> >
> > [3] Zhang, Ruiqi et al. “Negative Preference Optimization: From Catastrophic Collapse to Effective Unlearning.” ArXiv. 2024.
> >
> > [4] Fan, Chongyu, et al. "SalUn: Empowering Machine Unlearning via Gradient-Based Weight Saliency in Both Image Classification and Generation." ICLR. 2024.
> >
> > [5] Liu, Jiancheng, et al. "Model sparsity can simplify machine unlearning." NIPS. 2023.

---

> ### Comment · Reviewer_S1BA · 2024-08-11
>
> Thanks to authors for adequately addressing all the issues, therefore, I am raising my score to 7. Hope, you will find a space for the new experiments in the final version as they significantly bolster the paper.

---

> > ### Author Response · Authors · 2024-08-12
> > **Thank you!**
> >
> > We deeply appreciate your kind appraisal and positive feedback. We will carefully plan the final version of our paper to incorporate additional experiments.
> >
> > Thank you again for your valuable insights and dedicated effort in reviewing our manuscript.

---

### Official Review · Reviewer_fu6N · 2024-07-09

**Soundness:** 3
**Presentation:** 3
**Contribution:** 2
**Rating:** 5
**Confidence:** 3

**Summary:**

This paper summarizes the previous gradient-based unlearning methods and proposes three essential components for unlearning: weighted forgetting gradient ascent, remaining gradient descent, and a weight saliency matrix. Then, this paper derives the steepest descent direction and proposes a fast-slow weight method to update the model parameters. The experiments in both image classification and image generation shows the effectiveness of the proposed method.

**Strengths:**

1. This paper provides a new perspective to unify the gradient-based unlearning methods.
2. The experiments contain both image classification and image generation to show the effectiveness of the proposed methods
3. The derivation of the proposed method is attractive.

**Weaknesses:**

1. The gradient-based unlearning methods usually make more changes to the latter layers than the previous layers, which leads to inefficient information removal. This paper does not consider the drawbacks of such unlearning methods.
2. For the image classification task, this paper only tries the ResNet18 model. Then, for the image generation, this paper only tries DDPM. It might not be enough to claim to ''form a unified MU approach for all CV unlearning tasks, including image classification and image generation.''
3. This paper lacks more ablation studies to prove how SFR affect the unlearning results compared with other methods without the repairing stage on the updated model.
4. The main contributions of this paper are not clear. This paper claims to derive the steepest descent direction and proposes a fast-slow weight method. However, this paper still lacks a thorough advantage analysis of the steepest descent direction and the proposed update method.

**Questions:**

1. How many remaining data does the proposed method require, and how many remaining data is actually used in experiments?
2. The calculation of Hessian matrix $(H_t^r)^{-1}$ still demands large computational resources. Does the proposed method have any approximation on it?
3. Although this paper claims the proposed methods can significantly reduce the optimization steps in unlearning, why does the RTE not show significant reductions compared with SCRUB and SalUn?
4. In the implementations, $\nabla\mathcal{L}^r(\theta_*)$, $\nabla C(\theta_t)$, and $\nabla R(\theta_t)$ may not be zero. Thus, how large are the influences of the three approximation operations in eqs 3,4,6.

**Limitations:**

No further potential negative societal impact.

---

> ### Author Rebuttal · Authors · 2024-08-06
>
> Thank you for your valuable comments and noting that our method is attractive. We will address your concerns sequentially. **Relevant tables and figures are included in the attached rebuttal PDF**.
>
> **W1**: Addressing imbalance in layer-wise changes for enhanced information removal.
>
> > There are some arguments[1] regarding the parameter changes across different layers for various unlearning settings. We wish to clarify that our approach does not rely on heuristic strategies based on different layer-wise changes. Instead, our Balanced Saliency Mask can adaptively identify critical parameters for updating, aligning with SalUn[2] that has been verified effective for efficient information removal (Fig.R1). We hope that our theoretical foundation and approach could inspire further improvements to gradient-based unlearning.
>
>
> **W2**: Clarification on the range of models used in experiments.
>
> > There appears to be a misunderstanding regarding the diversity of models used in our experiments. We have explicitly demonstrated in the paper that for classification tasks, both **ResNet18** and **Swin-T** are employed (Tab.2). For generative tasks, our method extends beyond **DDPM** (Tab.3 & Fig.2), and also includes **Stable Diffusion**, which 'caters to practical scenarios' as mentioned by Reviewer HvBU (Tab.4). Moreover, we pioneer class-wise unlearning using **DiT** (Tab.3 & Fig.3). In response to the other two reviewers, we expand our experimental scope to include **large language models** in unlearning tasks (Review HvBU Q & Review S1BA W1) to further validate that our proposed method is **a unified MU approach for general setups**.
>
> **W3**: Ablation studies on SFR-on without repairing.
>
> > We conduct ablation experiments to assess the performance of our 'Sample-wise Adaptive Coefficient' and 'Balanced Weight Saliency' in the absence of repairing.
> >
> > The results in Tab.R2 affirm that these components alone can still enhance the baseline performance. We will include this ablation to support the effectiveness of our components.
>
> **W4**: Clarification on main contributions and advantage analysis.
>
> > We apologize for any lack of clarity in our presentation. We summarize our main contributions as follows:
> >
> > - We unify previous gradient-based unlearning methods from the perspective of the steepest gradient descent with three parts.
> > - We propose an improved approximate MU measured in a remain-preserving manifold.
> > - We introduce a fast-slow weight method to implicitly approximate the Hessian-modulated salient unlearning direction.
> >
> > Additionally, we would like to highlight the advantage analysis of our proposed method:
> >
> > - Following Proposition 2, our findings indicate that our method leverages retained curvature information and focuses more on forgetting, thereby ensuring efficient unlearning while preserving model performance.
> > - In 'Comparison with the joint loss' of Sec.4, we analysis the gradients of our method with those of the conventional joint loss and demonstrate that our method can prevent the forgetting direction from excessively affecting retained performance.
> >
> > Furthermore, we acknowledge in limitations (Appendix G) the absence of asymptotic analysis for our method on deep models. We look forward to future analyses that may inspire the community.
>
> **Q1**: Specification of remaining dataset size used in experiments.
>
> > In our formulation, we define the retain set as the complement of the forgetting set within the entire training dataset (Sec.2). For classification tasks, we set the retain set to align with our formalization, according to previous works[2]. However, as discussed in Sec.4 (lines 282-286), this may be impractical for generative tasks. Therefore, we randomly sample an equal number of samples from the remaining dataset as an insufficient retain set. Our experiments show that we still can achieve effective unlearning under such settings.
>
> **Q2**: Solution to the problem of demanding resources for calculating $(H_t^r)^{-1}$.
>
> > We agree with your concern regarding the substantial computational resources required to compute the Hessian inverse. To reduce this burden, we introduce fast-slow weight updates to directly approximate $(H_t^r)^{-1}\nabla\mathcal{L}^u(\theta_t)$, i.e., the Hessian-modulated unlearning direction (Proposition 3), which is actually one of our key contributions.
>
> **Q3**: Little reductions in RTE compared with baselines in the classification task.
>
> > In Tab.2, our method does not show significantly lower RTE compared to baseline methods because the efficiency improvement for well-studied image classification tasks is marginal. However, in the more challenging generative forgetting tasks, **our method exhibits a substantial reduction in RTE compared to SA and SalUn** (Tab.R4).
>
> **Q4**: The influences of approximation on practical implementations of Eqs.3,4,6.
>
> > We appreciate your attention to the influence of various approximations in practice. Next, we will illustrate these impacts on our propositions separately.
> >
> > * Eqs.3,4:
> >
> >    - Considering the approximations of $\nabla \mathcal{L}^r(\theta_0)$, $\nabla \mathcal{L}^r(\theta_*)$, and $\nabla R(\theta_t)$, we demonstrate in Fig.R2 their gradient norms under practical unlearning settings, which are consistent with our theoretical assumptions. In our method, $\nabla \mathcal{L}^r(\theta_0) \approx \nabla \mathcal{L}^r(\theta_*) \approx \nabla R(\theta_t) \approx 0$, allowing us to incorporate second-order information on the retain set.
>     - For $\nabla C(\theta_t)$, the derivative of the Euclidean distance metric in Eq.3 is generally non-zero, while the derivative of the output KL distance in Eq.4 is strict zero rather than approximation[3].
> >
> > * Eq.6:
> >    * We employ the proximal assumption of parameters near each updated model, verified by previous empirical observations[4].
> >
> > We will include these observations in our Appendix to bolster our analysis.

---

> > ### Comment · Reviewer_fu6N · 2024-08-13
> > **Thanks for your rebuttal.**
> >
> > Thanks for the authors' rebuttal. The rebuttal has solved my concerns on weaknesses 1, 3, and 4, and also on questions 2, 3, and 4.
> >
> > However, I still have the following three questions (not requiring extra experiments)
> >
> > 1. I acknowledge that currently, classification and generation are two major computer vision tasks for unlearning. Still, it does not mean all CV tasks consider the importance of image segmentation, object detection, etc. Although there are few works towards the unlearning on such tasks, they should also be considered if the paper wants to claim to work on all CV tasks.
> >
> > 2. Access to all the training data may not be practical, even in classification tasks. Then, I suggest that this paper can reduce the usage of training data or use a small public training data during unlearning, and it might also improve the time efficiency.

---

> > > ### Author Response · Authors · 2024-08-13
> > > **Response to additional questions**
> > >
> > > Additional **Q1**: Clarification on applicability to 'all CV unlearning tasks'
> > >
> > > > We apologize for any confusion caused by our unclear wording. As you have noticed that currently 'classification and generation are two major computer vision tasks for unlearning', our phrase 'all computer vision unlearning tasks' is intended to refer specifically to widely studied unlearning settings, including class-wise and random subset forgetting. We fully acknowledge that segmentation and object detection are also crucial and popular areas in CV, and MU should not exclude them. We believe that the progress of MU on classification and generation may spread to other CV tasks. **To avoid any future misunderstandings, we will refrain from making broad claims on 'all computer vision unlearning tasks' and instead use more precise and accurate descriptions of the task scope**.
> > > >
> > > > Furthermore, it is important to note that our method is not constrained by input modalities or loss forms. We are eager to explore the potential of our approach for unlearning tasks in image segmentation and object detection in future research.
> > >
> > > Additional **Q2**: Reducing the usage of training data in unlearning
> > >
> > > >We agree that employing limited training data presents a more realistic scenario, and we appreciate your insights on reducing training data usage. MU is still in its preliminary stages. Thus, the standard configuration of previous baseline methods uses all the remaining data. Our study adheres to this setting to ensure fair comparisons.
> > > >
> > > >Notably, our variant without using retain data (Tab.R2 in response to your W3) has outperformed previous baseline methods (Avg.D$\downarrow$: 2.37(SF) < 3.03(SalUn)). **This indicates that our proposed components are capable of efficiently forgetting even in scenarios devoid of retain data.**
> > > >
> > > > Your suggestion to reduce the training data also hits the essence: using retain data inherently involves a trade-off between remaining data performance and forgetting efficiency. We look forward to contributing to future research that aims to pursue better a trade-off with limited or even no retain data.
> > >
> > > Thank you for your detailed review and valuable comments. We notice that you mentioned there are three addition questions, but it appears we may have only received two. To ensure we can comprehensively address all concerns and refine our manuscript, could you please provide the details of the third question? Additionally, if there are any other concerns or suggestions that require further discussion, we warmly welcome your feedback!

---

> ### Author Response · Authors · 2024-08-06
> **Rebuttal Reference**
>
> > [1] Goel, Shashwat, et al. "Towards adversarial evaluations for inexact machine unlearning." ArXiv. 2022.
> >
> > [2] Fan, Chongyu, et al. "SalUn: Empowering Machine Unlearning via Gradient-Based Weight Saliency in Both Image Classification and Generation." ICLR. 2024.
> >
> > [3] Martens, James. "New insights and perspectives on the natural gradient method." JMLR. 2020.
> >
> > [4] Wu, Yichen, et al. "Meta Continual Learning Revisited: Implicitly Enhancing Online Hessian Approximation via Variance Reduction." ICLR. 2024.

---

> > ### Author Response · Authors · 2024-08-13
> > **Looking forward to further discussions (one day left before rebuttal ends)**
> >
> > We sincerely appreciate the time and effort you've invested in reviewing our manuscript. We hope that our responses have adequately addressed your concerns. As the discussion phase draws to a close in just one day, we kindly request your feedback on our rebuttal. If there are any remaining questions or concerns, we are looking forward to further discussions!

---

### Official Review · Reviewer_HvBU · 2024-07-12

**Soundness:** 3
**Presentation:** 3
**Contribution:** 3
**Rating:** 7
**Confidence:** 2

**Summary:**

This work introduces a perspective to unify previous machine unlearning approaches by decomposing the gradient descent direction into three components including forgetting gradient ascent, remaining gradient descent, and weight saliency matrix. The steepest descent direction is derived on the remain-preserved manifold, and a fast-slow weight method is proposed to implicitly approximate online Hessian-modulated salient forgetting updates.

**Strengths:**

1. This paper is well-written and well-structured. Each proposition is thoroughly proven in the appendix, and the preliminary section provides a detailed overview of the topic.

2. Experiments are comprehensive and encompass the most common settings in image classification and generation tasks. Additionally, the consideration of large pretrained models such as Stable Diffusion caters to practical scenarios.

**Weaknesses:**

I am not familiar with the concept of machine unlearning. It seems like this work is rigorous and comprehensive. From my perspective, there are no obvious weaknesses.

**Questions:**

Is the proposed method available for models in other modalities, apart from images? Including experiments on other modalities such as large language models could further broaden the scope of this work.

**Limitations:**

Please refer to the "Weaknesses" and "Questions" sections.

---

> ### Author Rebuttal · Authors · 2024-08-06
>
> Thank you for your positive feedback and valuable comments on our presentation and extensive experiments. In Appendix B, we have included **Related Works** to help readers understand the concept of Machine Unlearning and to introduce some previous unlearning methods. **The relevant Tab.R1 is included in the attached rebuttal PDF**. Below, we address your request to add experiments on large language models (LLMs):
>
> **Q**: Adding experiments on more modalities such as LLMs.
>
> > As stated in Sec.4, our analysis and method are not limited by the model input modality. Therefore, **our method can be seamlessly extended to other modalities beyond images**, such as LLMs, to achieve efficient forgetting.
> >
> > We conduct experiments using the recently proposed benchmark of TOFU[1] fine-tuned Phi-1.5[2] to evaluate the effectiveness of our method in the LLM unlearning task, compared with four LLM unlearning baselines: gradient descent(GA), gradient difference(GradDiff[3]), negative preference optimization(NPO[4]), and its enhanced version. The TOFU dataset comprises fictional author biographies, along with questions and answers related to the authors' attributes, which helps assess methods of data forgetting on fine-tuned LLMs.
> >
> >  As shown in Tab.R1, our method achieves superior forgetting quality, making the unlearned model most indistinguishable from the retrained model based on the Truth Ratio distribution of the forget set. Additionally, our method efficiently maintains the model utility.
> >
> > We greatly appreciate your suggestion to include experiments on modalities beyond images. We will incorporate this experiment and details of the LLM unlearning task in our Appendix to broaden the scope of our work.
>
>
> **Reference**:
> > [1] Maini, Pratyush et al. “TOFU: A Task of Fictitious Unlearning for LLMs.” ArXiv. 2024.
> >
> > [2] Li, Yuan-Fang et al. “Textbooks Are All You Need II: phi-1.5 technical report.” ArXiv. 2023.
> >
> > [3] Liu, B. et al. “Continual Learning and Private Unlearning.” ArXiv. 2022.
> >
> > [4] Zhang, Ruiqi et al. “Negative Preference Optimization: From Catastrophic Collapse to Effective Unlearning.” ArXiv. 2024.

---

> > ### Comment · Reviewer_HvBU · 2024-08-12
> >
> > I thank the authors for their response. I've read the related work section in Appendix B to learn more about the concept of Machine Unlearning, and I appreciate that the authors have included additional experimental results in the attached rebuttal PDF. I have accordingly increased my confidence score.

---

> > > ### Author Response · Authors · 2024-08-13
> > > **Thank you!**
> > >
> > > We deeply appreciate your kind appraisal and positive feedback. We will incorporate additional experiments on NLP in our final version.
> > >
> > > Thank you again for your helpful comments and dedicated effort in reviewing our manuscript.

---

### Author Rebuttal · Authors · 2024-08-06

# General Response
Dear Program Chairs, Area Chairs, and Reviewers,

We sincerely appreciate your time, constructive critiques, highly pertinent concerns, and valuable suggestions, all of which substantially help improve our work. We are also grateful to the reviewers' consistent acknowledgment of our rigorous mathematical framework, the performance and efficiency of our method, and the comprehensive evaluation across various experimental settings. We address all questions point by point, with **supporting tables and figures included in the rebuttal PDF**. Here is a brief summary:

**Additional Experiments**:
* We extend our method to large language model (LLM) unlearning tasks, demonstrating superior performance compared to existing baselines, thereby broadening the applicability of our approach. (Reviewer HvBU Q and Reviewer S1BA W1)
* Proportion of parameters activated by the saliency mask to balance layer-wise change for efficient forgetting. (Reviewer fu6N W1)
* Ablation studies on the absence of repairing operations. (Reviewer fu6N W3)
* Assessment of the impact of approximation in practical unlearning process. (Reviewer fu6N Q4)
* Unlearning at the granularity of a single data point or other suitable levels. (Reviewer S1BA Q1)
* Running-time efficiency analysis. (Reviewer fu6N Q3 and Reviewer S1BA Q3)

**Clarification**:
* Comprehensive range of models and settings employed in our experiments. (Reviewer fu6N W2)
* Advantage analysis of our proposed approximate MU in the remain-preserving manifold and fast-slow weight update method. (Reviewer fu6N W4)
* Remaining dataset size used in experiments. (Reviewer fu6N Q1)
* Approximation to solve the problem of unaffordable cost in calculating the Hessian inversion. (Reviewer fu6N Q2)
* Unlearning metric on accuracy disparity. (Reviewer S1BA W2 and Q2)

---

### Decision · Program_Chairs · 2024-09-25

**Decision:**

Accept (spotlight)

**Comment:**

The paper studies an important problem of machine unlearning, i.e. forgetting the influence of some data points, while preserving the remain geometry. The paper introduces a perspective to unify previous machine unlearning approaches by decomposing the gradient descent direction into three components: forgetting gradient ascent, remaining gradient descent, and weight saliency matrix. Then the steepest descent direction is derived based on the remain-preserving manifold geometry. Moreover, to overcome the computationally-intensive inverse Hessian calculation, a fast-slow weight method is proposed to implicitly approximate the Hessian-adjusted direction. The efficacy of the proposed methodology is demonstrated by some image classification (ResNet and DiT), image generation by DDPM, and LLM as well during rebuttal.

After rebuttal, all the reviewers agree to accept the paper, so is the final decision, provided that the authors could reflect those discussions in the final version properly.